# Coastal Boulder Dynamics Inferred from Multi-Temporal Satellite Imagery, Geological and Meteorological Investigations in Southern Apulia, Italy

Marco Delle Rose [1,*] , Paolo Martano [1] and Luca Orlanducci [2]

1  National Research Council of Italy, Institute of Atmospheric Sciences and Climate, 73100 Lecce, Italy; p.martano@isac.cnr.it
2  Geopro, Geological Prospections, 73100 Lecce, Italy; lucaorlanducci@libero.it
*  Correspondence: m.dellerose@le.isac.cnr.it

**Abstract:** Boulder dynamics may provide essential data for coastal evolution and hazards assessment and can be focused as a proxy for the onshore effect of intense storm waves. In this work, detailed observations of currently available satellite imagery of the Earth surface allowed us to identify several coastal boulders displacements in the Southern Apulia coast (Italy) for a period between July 2018 and June 2020. Field surveys confirmed the displacements of several dozens of boulders up to several meters in size, and allowed us to identify the initial position for many of them. Two possible causative storms were identified analysing archive weather maps, and calculations based on analytical equations were found in agreement with the displacement by storm waves for most of the observed boulders. The results help to provide insights about the onshore effect of storm waves on the coastal hydrodynamics and the possible future flooding hazard in the studied coast.

**Keywords:** marine weather; characteristic wave height; storm surge; shore platform; overtopping wave; hydrodynamics equation; flooding hazard; open source satellite image

## 1. Introduction

Boulder dynamics is an issue of growing concern in Earth Sciences since it may provide essential data for coastal evolution and hazards assessment [1–5]. Geomorphological monitoring is the primary tool to study the coastal morphodynamic processes, and particularly boulder transport [6,7]. Several monitoring methods have been recently used, including terrestrial laser scanning, drone photogrammetric survey, transect photo sets, kite aerial photography, surveillance camera recording, and radiofrequency identification [8–13]. However, due to budget and organizational reasons, they are used to monitor short stretches of coast (hundreds of meters long).

In the case of clasts of large size, multi-temporal satellite image analysis can help to overcome such restrictions [14,15], including those made with free computer programs, such as Google Earth (GE) [16,17]. The identification of displaced boulders can then be followed by on site surveys on the selected places. While careful geological and geomorphological investigations are mandatory to improve our sedimentological and morphodynamic knowledge [18–20], basic geometrical, physical, and kinematic features of boulders displaced during high energy events give a measure of the wave impact on the coast [21–23].

The potential of storm waves to move boulders and contribute to the coastal morphological evolution is of increasing interest also in connection with possible changes in the storm climatology related to climate change, with a possible increase in their energy release [24,25]. The present study obtains information about the dynamics of the boulders that occurred between two consecutive image sets available on GE. The study area is the southern coast of Apulia (Italy, central Mediterranean). It is known in the literature for its widespread boulder fields, whose causative events are traced back to both tsunamis and storms [1,26–28].

The aims of the present study are: (a) to provide data on boulder displacements for a wide study area by using publicly available satellite imagery coupled with field surveys; (b) to identify the storm event(s) resulting in boulder displacements; (c) to use boulders as a proxy for coastal hazards in a reliable way. GE provides several open source tools for Earth observation studies [29,30]. The use of such a facility in coastal boulder research has recently started, even if its potential has yet to be fully understood [16,17].

In the present paper, first results were gained for the Apulia coast as exposed in the next sections. Details and data supplemental to the main text are given in four appendices: in Appendix A geomorphological features of main identified boulders are reported; in Appendix B, an example of a measurement procedure is shown; in Appendix C, some weather maps of the meteorological features are shown; and finally, in Appendix D, hydrodynamics equations are briefly summarized, together with the calculated minimum wave heights for the boulder displacements.

The results found a global agreement between the boulder transport distance, as detected by satellite imagery and field geological surveys, and the characteristic wave heights of the strongest identified storms, thus, giving support to use the displaced boulders as proxies for the coastal flooding hazard.

## 2. Study Area

The Southern Apulia (also named Salento Peninsula) belongs to the Apulia Carbonate Ridge, a foreland sector of the Apenninic Chain. It developed a horst and graben setting from the Late Cretaceous, undergoing mild structural deformations [31]. The upper basement of the Salento Peninsula consists of Mesozoic limestones and dolostones covered by Tertiary and Quaternary carbonates and marls (Figure 1).

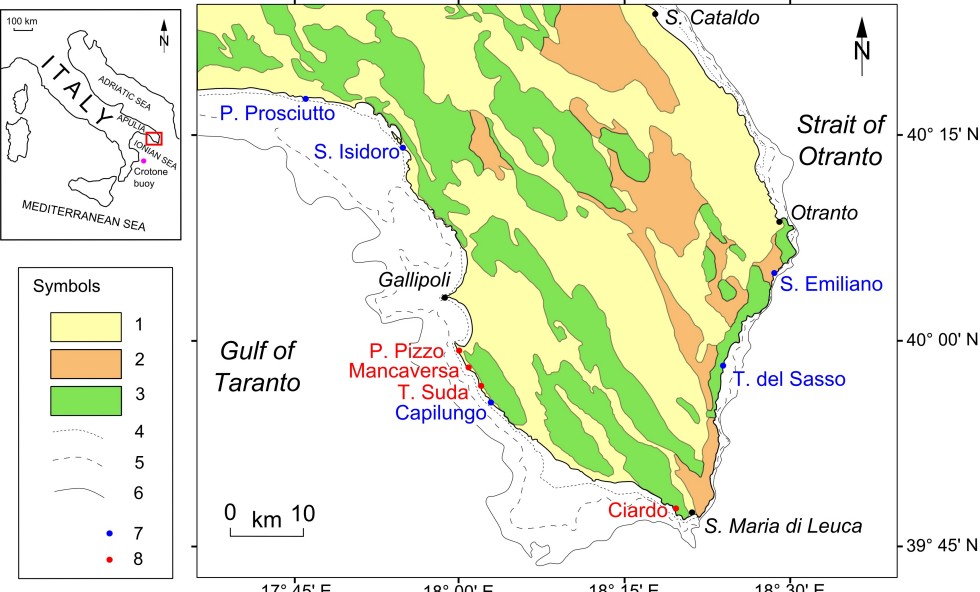

**Figure 1.** Simplified geological map of Southern Apulia: 1, Quaternary units (carbonates, marls, and bioclastic deposits); 2, Tertiary units (carbonates and marls); 3, Cretaceous units (limestones and dolostones). Other Symbols: 4, −10 m isobath; 5, −20 m isobath; 6, −50 m isobath; 7, previously known site; 8, new site. Isobaths from the nautical chart of "Istituto Idrografico della Marina Militare" of Italy (1:100,000).

Horst and graben are dissected by normal or strike-slip faults. Due to the predominance of calcareous rocks, the study area is typified by karst features. The younger phase of karstification started from the Early Pleistocene with the tectonic uplift of the Apulia Carbonate Ridge. The concomitant marine regression was discontinuous and affected by eustatic processes, thus, resulting in a marine terrace staircase [32,33]. At each of the coastal

stretch herein considered, the lower marine terrace has peculiar stratigraphic and geomorphological characteristics, especially with regards to the width of the coastal platform and the height of the cliff edge.

The coast of the southern Apulia faces the northern Ionian Sea and has a microtidal regime. The north-western side of the Ionian Sea coincides with the Taranto Gulf, a semi-enclosed basin. Toward the north-east, the Ionian Sea is connected to the Adriatic Sea by the Strait of Otranto. The physiography and coastal morphology west and east of S. Maria di Leuca (Figure 1) are different. The south-western coast of Salento Peninsula is characterised by a series of pocket-beaches and rocky cliffs. The edge of these latter is usually between 0.5 m and a few meters above Mean Sea Level (MSL), while their base is located a few meters below. The shoreface is gently sloping, with the 10 m isobath typically tens of meters away from the coast (data from the nautical chart of "Istituto Idrografico della Marina Militare" of Italy).

By contrast, an escarpment up to tens of meters high bounds the Salento Peninsula toward the Otranto Strait. It abruptly slopes to the sea and usually dips at relatively high angle down to about 50 m in depth [34]. The escarpment is lower at some stretches, as the one south of Sant'Emiliano (S. Emiliano in Figure 1). The sea-state reference for the northern Ionian Sea is the Crotone buoy (inset top left of Figure 1). It belongs to the Ondametric Network of Italy. By analysing the datasets of this buoy, maximum significant waves of 6.3 m for a 50 year return period and of 8.2 m for a 100 year return period have resulted [35,36].

## 3. Methods

### 3.1. Satellite Imagery and Geological Investigations

The most recent open source images of the Southern Apulia, available on GE from December 2020, dated on 28 June 2020. Previous images were taken on 20 July 2018, 9 July 2017, and 19 July 2015. As a first step, the rocky coastal stretches were carefully examined at an eye elevation of between 50 and 150 m. On each boulder field and single boulder with major axis of at least 1 m, visual comparisons were made between the last satellite image and the previous one. The main goal was to establish any changes in position [4,17,37]. All the displaced boulders identified using GE images were then surveyed on the field. Any uncertainties were resolved by feedback control (Figure 2).

Three different types of initial positions were dealt with: (a) socket; (b) print left on the platform by singular boulder; (c) print left by clustered boulder. Since a socket is defined as the detachment surface of the boulder from the rocky substrate [3], the difference in colours between the fresh (not or poorly covered by lichen) and the weathered rocky surface usually allow easy detection [7]. Sometimes in satellite images, the socket is highlighted by the shadows of its edge [38].

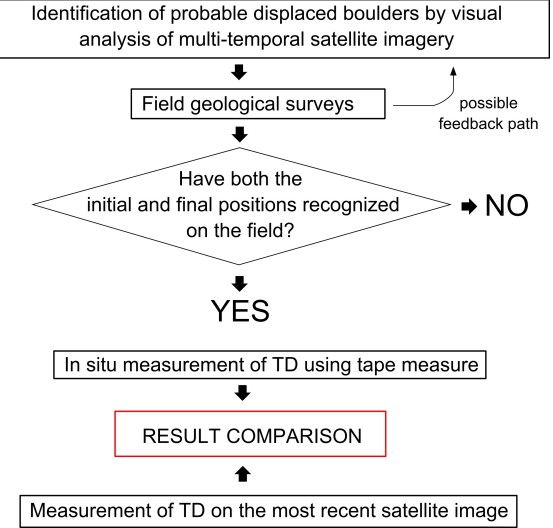

**Figure 2.** Flowchart explaining TD measuring and testing.

The case of boulders previously uprooted from their original substrate is different. The prints left on the surface by such boulders are more or less evident, and, from post-displacement satellite images, it is possible to identify them in favorable conditions of sunlight. From pre-displacement satellite images, such initial positions must be clear and unambiguous in order to be considered [39]. However, the check by field survey is crucial, which, in the case of single boulders, may allow to establish its initial position with extreme accuracy. The initial positions of close or overlapping boulders may be determined as the area occupied by the whole group, rather than that of any single boulder.

The in situ determined data were: dimensions of the boulder axes, $a$ (major), $b$ (middle), and $c$ (minor) (according to [40]); Initial Distances from the cliff edge, $x_i$; Final Distances from the cliff edge, $x_f$; Pre Transport Setting (PTS); Joint-Bounded (JB); Sub-Aerial (SA); SubMerged (SM) (according to [21]); Movement Type (MT); saltation (ST); sliding (SL); overturning (OV); Lithology (Li); Calcarenite (C); Limestone (L); and Sandstone (S).

Some geomorphological observations were made in view of further detailed studies. Where both the initial and the final positioning of the boulders were identified, the Transport Distance (TD) was determined (Figure 2, see also Appendix B). The axes dimensions were used to establish the Size (Si) and Shape (Sh) of the boulders according to [41] as well as the Flatness Index (FI) = $(a + b)/2c$. The data collected by satellite imagery and geological investigations are shown in Section 4 and Appendix A.

*3.2. Marine Weather Analysis*

A visual screening of the Mediterranean storms in the time interval of this study has been made using the web archives of the GLOBO-BOLAM-MOLOCH model cascade [42]. The first analysis was made by a careful screening of the 500 hPa geopotential height maps of the GLOBO model archive for the Northern emisphere. These identify the middle-atmosphere pressure conditions that allow to ensure or exclude the presence of stormy conditions over the Mediterranean region.

After this coarse-grained identification of the storm periods, these were analysed in higher space-time resolution using the archives of the mesoscale models BOLAM (10 km horizontal resolution) and MOLOCH (3 km horizontal resolution), in order to obtain more detailed information about the wind field over the study region. Indeed, the evaluation of the characteristic wave height $H_0$ and the wave period $T$ in the sea storms was made starting from the wind characteristics (wind speed $U$, wind direction, fetch over the sea surface $F$, and duration $R$) as established by the forecast map archive of the BOLAM-MOLOCH model.

The wind characteristics were used to evaluate the spectral peak wave height from the similarity relation equations [43,44]. They related $T$, $H_0$ and $R$ to the wind speed at 10 m height offshore $U$ and to the fetch over the sea $F$:

$$gT/U = 0.286(gF/U^2)^{1/3} \tag{1}$$

$$gH_0/U^2 = 0.0016(gF/U^2)^{1/2} \tag{2}$$

$$gR/U = 68.8(gF/U^2)^{2/3} \tag{3}$$

where $g$ is the gravity acceleration ([44], p. 117).

The duration $R$ of stability of wind conditions, estimated from the wind maps, has been used as a limiting factor to calculate an effective fetch $F$ for the Equations (1) and (2), whenever the actual geometric fetch appears to be much longer.

The sea level elevation $H_s$ related to the storm surge over the south-western Salento coast can be estimated by a simple analytical model depending on the wind stress along the coast and the depth of the coastal shelf. For a storm surge depending on the blocking effect of an open coast over an Ekman current caused by a transient wind stress parallel to the coast line, the following expression can be found ([45], p. 396):

$$H_s = U*^2 C^{-1} rt\textbf{exp}(-Lf/C) \tag{4}$$

with $C = (g\,D)^{1/2}$; where $D$ is the coastal shelf depth, $L$ is the length of the coastline along the right side of the wind direction, $f$ is the Coriolis parameter ($f = 2\Omega\sin\varphi$, $\Omega$ = earth angular velocity, $\varphi$ = latitude), $r$ is the ratio of the specific weight of air and water, and $t$ is the transient time of the storm. $U*$ is the friction velocity, which can be calculated from the wind speed $U$ at the height z = 10 m over the sea level solving the following equation ([44], pp. 115–116):

$$U(z) = (U*/k)\ln(zg/(aU*^2))\tag{5}$$

Here, $k = 0.4$ is the Von Karman constant, and the Charnock parameter has an average value $a = 0.018$; however, it appears to vary with sea conditions and tends to increase over young sea as during the wave growth in storms. Hsu ([44], pp. 116–117), proposed $a = (2\pi/g)(H_0/T^2)$ to take into account this effect. Empirical formulas relating the sea level to the depth of the surface low pressure centre in the storm are also used in practical estimations of the storm surge, although they are dependent on the site of application:

$$H_s = S F_s F_m\tag{6}$$

where the open sea surge $S$, the shoaling factor $F_s$, and the storm motion factor $F_m$ are tabulated from empirical observations ([44], pp. 212–215). The open sea surge depends on the depth of the low pressure centre, that, together with the average speed and direction of the storm motion during 24 h can be estimated by the BOLAM forecast maps, while the shoaling factor is tabulated as a function of the depth of the coast shelf. The marine weather conditions during the considered storms are exposed in Section 5 and Appendix C.

### 3.3. Onshore Wave Height Assessment

Coastal boulders can be used as proxies to estimate coastal hazards, among which extreme wave inundations and damaging flows [5,46,47]. The hydrodynamics equations [21,48], largely applied to calculate the minimum wave height $H_m$ to move the boulders, however, have been criticized for several shortcomings, and care should be used in drawing conclusions [49]. In addition, where the initial position of the boulders may be recognized, the onshore decay of the wave height should also be considered [49–51].

As shown in Section 4 and Appendix A, the surveys made at the boulder sites generally allowed us to determine the initial positions. Thus, to estimate the Sea Wave Height (SWH) in each determined onshore point, the equation proposed by [50] was (Section 6 and Appendix D). This allowed us to calculate the SWH reduction over the shore, which is an estimation of the wave height $H$ impacting the boulder at a distance $x_i$ from the shoreline.

The obtained estimate of the height of the wave impacting the boulder, for each of the selected storms, were then compared with the required minimum wave height $H_m$ to move the boulders according to the observations of their position and size. In the comparison, to calculate $H_m$, the dynamic equations by Nandasena et al. (2011) [21] for the initial position and Engel and May (2012) [48] for the final position were used (see also Appendix D).

## 4. Satellite Imagery and Geological Investigations

### 4.1. Overview

The first achievement of this study is the identification of 81 boulders displaced along eight coastal stretches between July 2018 and June 2020 (Table 1). Four of these eight sites (Punta Prosciutto headland, north of Torre Sant'Isidoro, Capilungo coast, south of Torre Sant'Emiliano; P. Prosciutto, S. Isidoro, Capilungo, and S. Emiliano in Figure 1) are already known in the boulder literature [10,26–28]. At Torre del Sasso (T. del Sasso in Figure 1), another previously known site, no changes apparently occurred in the positions of coastal boulders during the considered time interval.

**Table 1.** Acronym code (ID Code), number of identified Boulders, and number of measured Transport Distances (No. of TD).

| Site Name | ID Code | Boulders | No. of TD |
|---|---|---|---|
| Punta Prosciutto | PR | 18 | 13 |
| Sant'Isidoro | SI | 3 | 2 |
| Punta Pizzo | PI | 18 | 13 |
| Mancaversa | MA | 15 | 12 |
| Torre Suda | SU | 14 | 9 |
| Capilungo | CA | 6 | 6 |
| Ciardo | CI | 5 | 4 |
| Sant'Emiliano | SE | 2 | - |

Many of displaced boulders (47 out of 81) were identified at the 6.5 km long stretch of coast between Punta Pizzo headland and Torre Suda. This coast includes three of the new identified sites (P. Pizzo, Mancaversa, and T. Suda sites of Figure 1). The Torre Suda site only was previously known for the boulder dynamics from two preliminary reports [52,53]. This should instead be the first report for the Ciardo site (west of S. Maria di Leuca, Figure 1).

The initial and final coordinates of the displaced boulders taken from the 28 June 2020 GE image are in Tables A1, A3, A5, A7, A9, A11, A13, A15. For 59 out of 81 boulders, the TD were calculated with varying degrees of accuracy, as explained below; in column 4 of Table 1 the number of such cases (No. of TD) for each site is reported. The Pre Transport Setting (PTS) was determined for all the 81 boulders, while the Movement Type (MT) was indeterminable for 12 boulders, uncertain for 11, certain for the remaining 58 (Tables A2, A4, A6, A8, A10, A12, A14 and A16). The main features to each site are provided below, while further field data are in Appendix A.

### 4.2. Punta Prosciutto Headland

Punta Prosciutto headland is placed at the north-western side of the study area (Figure 1). The average height above the MSL ($H_c$) of the cliff is about 0.5 m, while the shore platform is made by a calcarenitic terrace lying on the not outcropping Cretaceous limestone. By comparing the satellite images available on GE, 18 boulders displaced by waves between July 2018 and June 2020 have been identified (Tables A1 and A2). Most of the boulders are scattered on the west side of Punta Prosciutto headland (Figure 3).

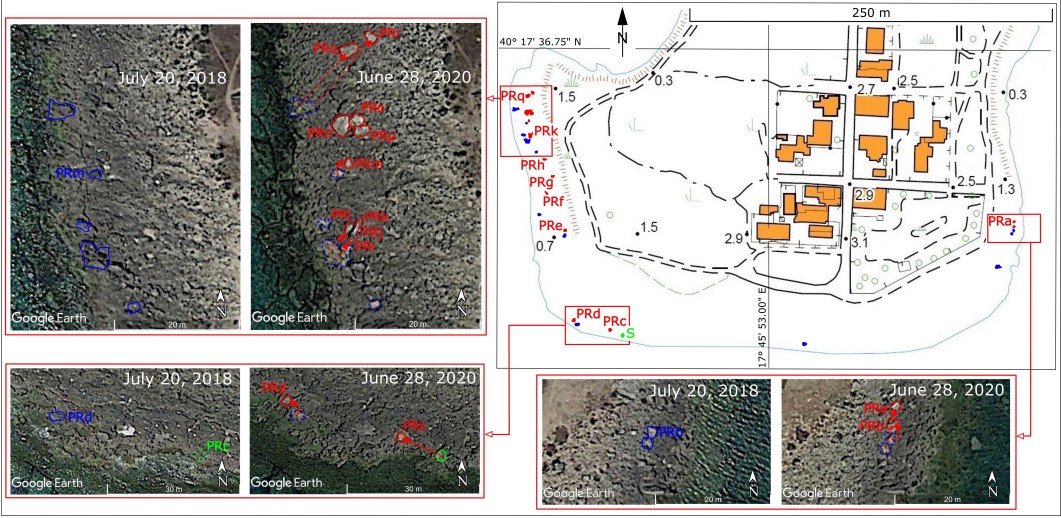

**Figure 3.** Comparative satellite images of Punta Prosciutto. The initial positions of the boulders are highlighted in blue, the final ones in red, and sockets are green. Red arrows highlight the displacements. Topography from the technical map of Lecce Province (1:5000).

Eleven of the eighteen boulders are isolated, while the others are clustered as two groups (Figure 4). The first is composed of four imbricated boulders (PRi,j,k,l); the second, by three neighbouring boulders (PRn,o,p). The initial positions of the first group were identified, albeit not individually (in Italics in Table A1; see the blue lines in Figure 3), while no evidence of the initial positions of the boulders of the second group was found. The direction of displacement is generally from SW to NE, except for the boulders PRc,d for which it is from SE to NW (Figure 3). The 13 calculated TD range from 2.6 to 16.3 m (Table A1); the magnitudes of the large boulders PRf, PRq, and PRr (Table A2) are of 16.3, 11.7, and 15.4 m, respectively. The field evidence shows that PRc moved out of its socket, while the other boulders had previously been detached from the substrate.

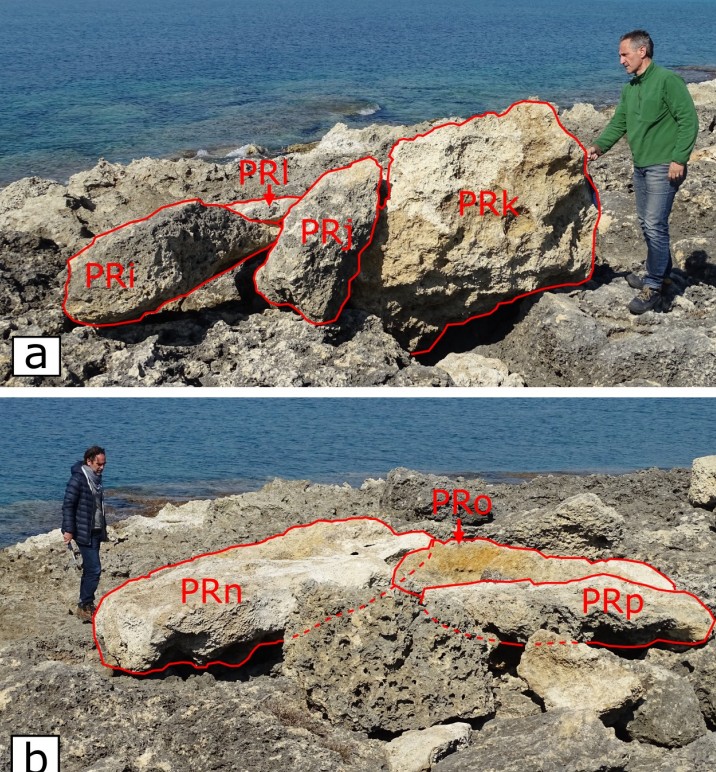

**Figure 4.** Two main boulder clusters of Punta Prosciutto headland: (**a**) Four imbricated boulders arranged in the SW-NE direction. (**b**) Group of closely spaced boulders that includes the largest (PRn) of this stretch of coast.

### 4.3. Sant'Isidoro Coast

North of Torre Sant'Isidoro coast (S. Isidoro in Figure 2) is monitored since 2017. By means of such field activity, 11 boulders were identified as being displaced by the storm of 12–13 November 2019 [10]. However, due to their small sizes, only three of these boulders were recognized by comparing GE satellite images (Tables A3 and A4). In addition, the sockets are not visible by satellite images (Figure A1). The largest dislodged boulder (4.8 × 2.2 × 1.1 m) is not recognizable by the used satellite imagery due to its very short TD (0.2 m; cf. [10] also for other features of the site). Since the November 2019 storm and until June 2020, there have been no further changes in the position of the boulders. Cretaceous limestone crops out over much of this site, however the 12–13 November 2019 storm produced only calcarenitic boulders. The average height above the MSL ($H_c$) of Sant'Isidoro cliff is less than 0.5 m.

### 4.4. Punta Pizzo Headland

The coast south of Punta Pizzo Headland (P.Pizzo in Figure 2) is roughly straight and bounded by a cliff about 1.5 m high above the MSL ($H_c$). A total of 18 boulders have been identified at this site, grouped into three different areas (Figure 5).

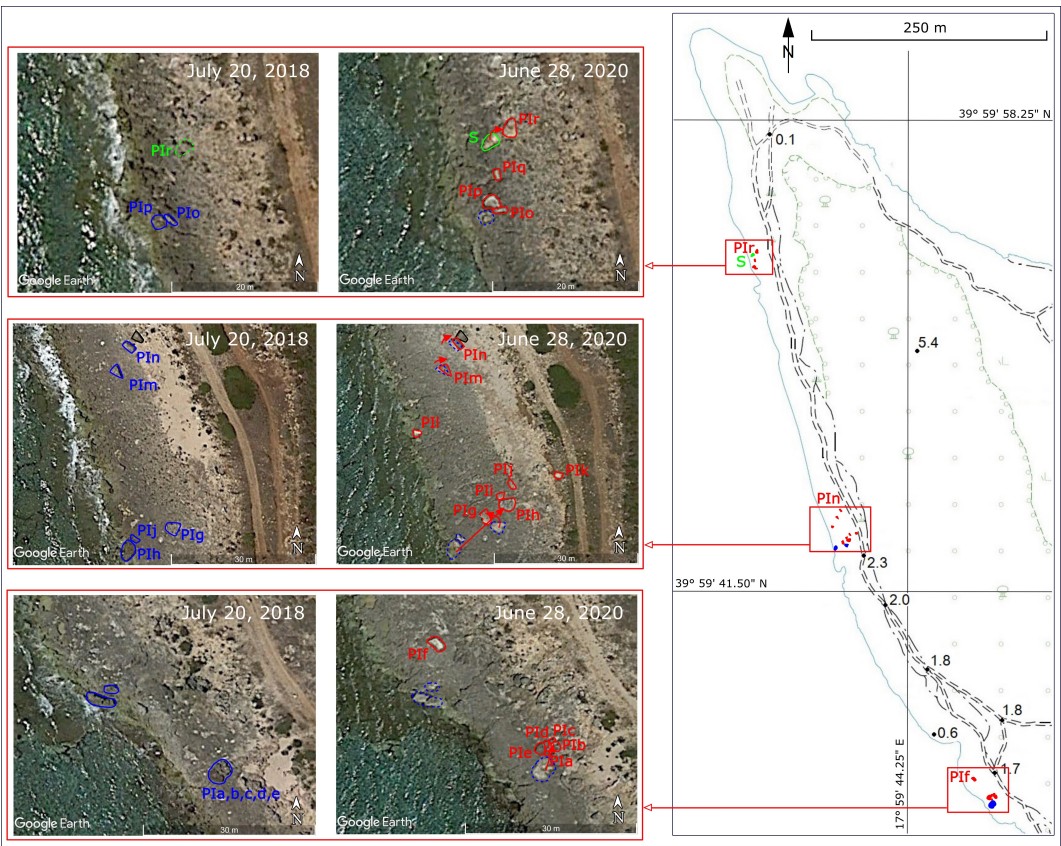

**Figure 5.** Comparative satellite images of Punta Pizzo. The initial positions of the boulders are highlighted in blue, the final ones in red, and the sockets are green. Red arrows highlight the displacements. Topography from the technical map of Lecce Province (1:5000).

The northern area includes four boulders that changed position in the period considered in the present study. Regarding the overturned boulder, PIr, its socket has been also recognized (Figure A2a,b). The central area has the largest number of boulders, including the largest one—namely the boulder PIh (Table A6). For this latter, a TD of 14.3 m was measured, lower only than that of the smaller boulder PIj (Table A5). To give an example of the used procedure, TD measures of the central area are shown in Appendix B (see also Figure 2).

In the southern area a group of five closely spaced boulders (PIa,b,c,d,e) is to be emphasized (Figure A2c,d). Each of these boulders has been moved about 4–5.5 m from the initial positions, clearly visible in the GE image of July 2018 as well as by the prints left on the platform and visible in the satellite image of June 2020 (Figure 5). The direction of displacement is from SW to NE for all boulders whose initial and final positions have been recognized along the coast of Punta Pizzo headland.

### 4.5. Mancaversa Coast

The average height above the MSL ($H_c$) of Mancaversa cliff does not exceed 1.5 m. A particular feature of this stretch of coast is the presence of sandstone dunes that widely cover the calcarenitic terrace. Two of the fifteen boulders on this site are in fact made of sandstone, the others of calcarenite (Tables A7 and A8). The fifteen boulders are grouped into four different areas (Figure 6, Appendix A). The southernmost boulder, MAa, is the

largest in this site (Figure A3a). Its initial position can be defined with some difficulty based on the satellite image of July 2018; however, it is clearly visible in the image of July 2017.

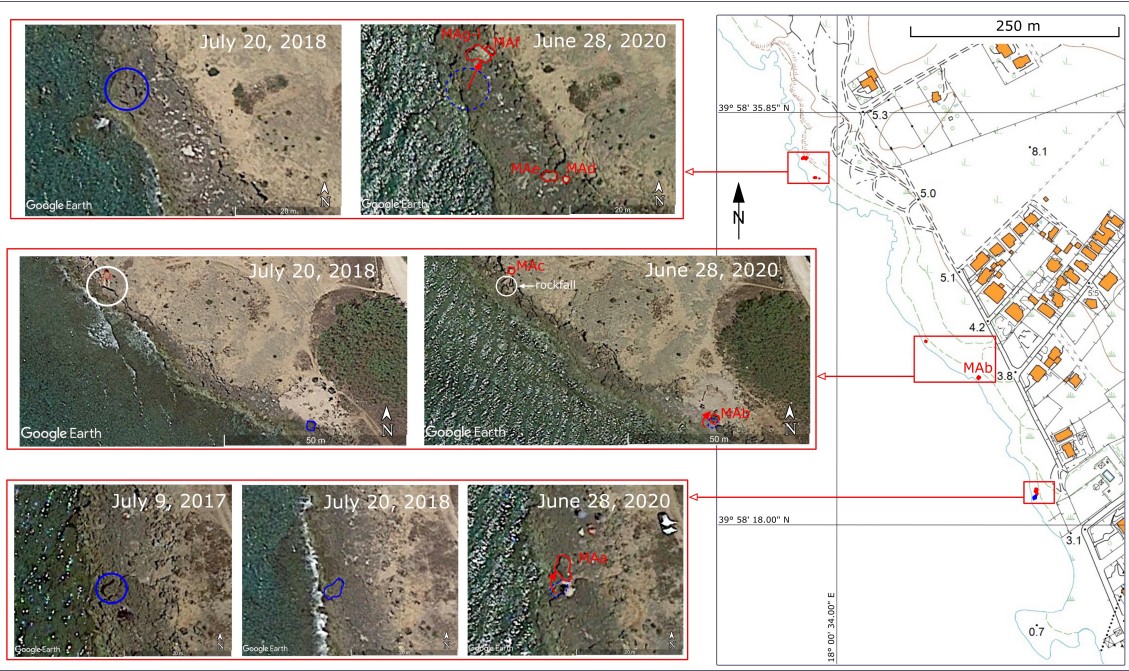

**Figure 6.** Comparative satellite images of Mancaversa. The initial positions of the boulders are highlighted in blue, the final ones in red; a rockfall are white. Red arrows highlight the displacements. Topography from the technical map of Lecce Province (1:5000).

The group of seven imbricated boulders MAf-l must be mentioned (Figure A3b). As for the case described in the previous subsection, the initial positions are clearly visible in the Google Earth image of July 2018. Instead, the prints left on the platform are hardly visible in the June 2020 GE image, and even the geological field observations have not provided good results in this regard. The alteration of the rock has evidently already hid this feature. It was however calculated that each of the seven boulders has been transported about 5.5–8.5 m from the initial positions (Table A7). The rockfall next to the sandstone boulder MAc (Figures 6 and A3c) may have been triggered by the impact of storm waves.

*4.6. Torre Suda Coast*

By comparing the GE satellite images, fourteen displaced boulders were identified at the Torre Suda (Tables A9 and A10, Figure 7). They are grouped into four different areas (Appendix A). The northern area was geologically surveyed after the 29–30 October 2018 storm because of a first displacement of the boulder SUi [52]. This circumstance makes it possible to have further data on the reconstruction of the boulder dynamics occurred in the time period considered by this study. SUa (Figures 8a and A4) is the largest boulder among those identified in all the sites considered herein. Its initial and final positions are clearly visible in the satellite images. Boulder SUa underwent a transport parallel to the coast of about 9 m (from SSE to NNW), without going beyond the cliff, which is about 0.5 m high ($H_c$), even if it exceeds 1 m for short distances. Thus, despite having a relevant TD, its distance from the cliff edge ($x_f$) has not changed.

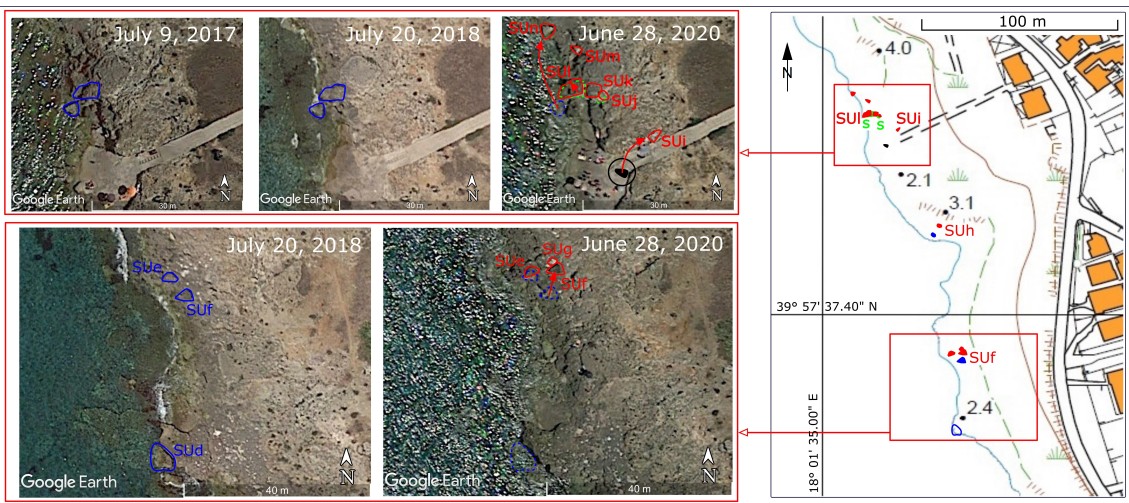

**Figure 7.** Comparative satellite images of Torre Suda. The initial positions of the boulders are highlighted in blue, the final ones in red, and the sockets are green. Red arrows highlight the displacements. In the June 2020 image, the boulder SUi as transported by the October 2018 storm is reported (in dark). Topography from the technical map of Lecce Province (1:5000).

As mentioned above, the boulder SUi suffered a first transport during the October 2018 storm from SW to NE. Subsequently, it was further transported towards the hinterland in the same direction, as determined by the GE image and field survey. The boulders SUj,k,l,m,n are scattered in the northernmost area of the Torre Suda site (Figures 7 and 8b). The aforementioned storm is excluded as the causative event of the change in their positions.

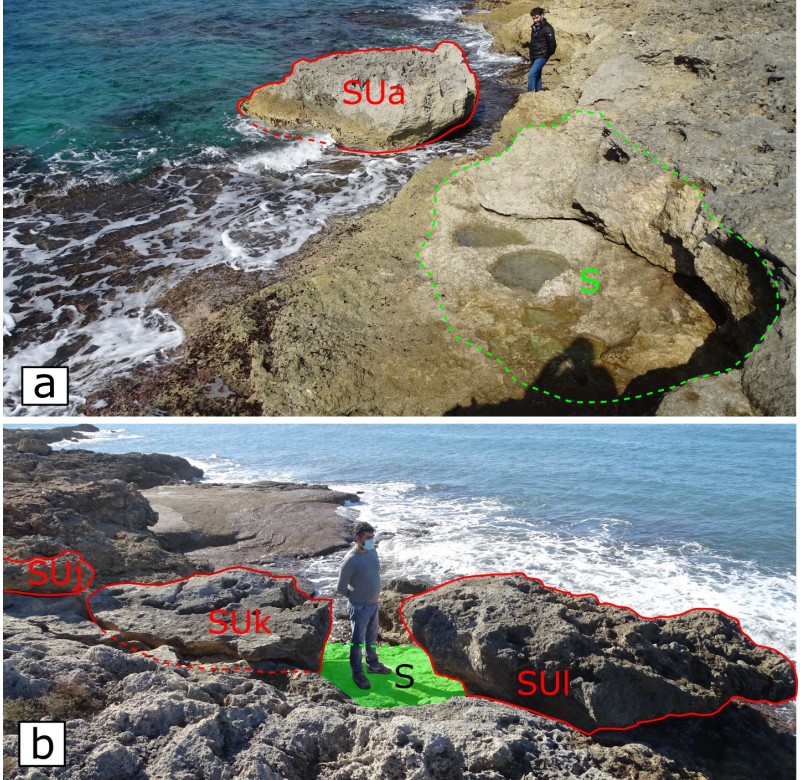

**Figure 8.** Torre Suda coastal stretch: (**a**) the largest boulder SUa and its socket taken from S. (**b**) The boulders SUj,k,l taken from N; the portion of SUl's socket exposed by the displacement is highlighted in green.

### 4.7. Capilungo Coast

The average height above the MSL ($H_c$) of Capilungo cliff does not exceed 0.5 m. Six boulders changed position between July 2018 and June 2020 in this stretch of coast (Figure 9, Tables A11 and A12). CAa,b,c form a group of closely spaced boulders and have undergone a transport (TD) of 2–3 m (Figure A6). The other boulders of this site have undergone shorter transports but still more than 1.5 m.

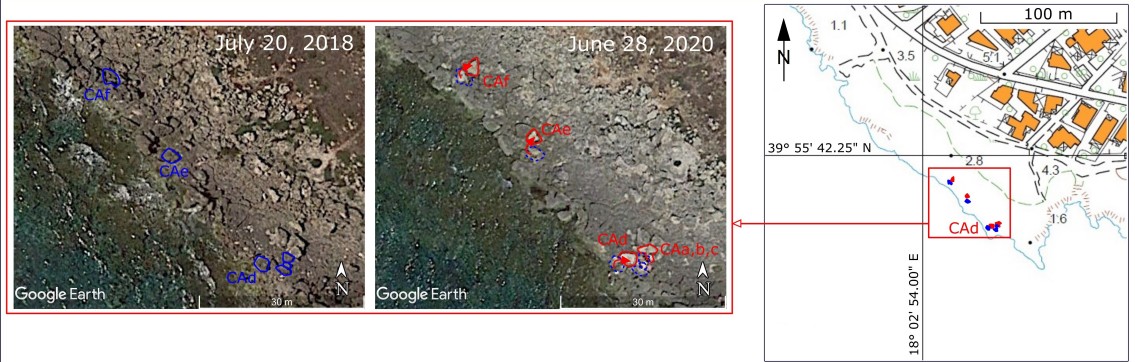

**Figure 9.** Comparative satellite images of Capilungo. The initial positions of the boulders are highlighted in blue, and the final ones in red. Red arrows highlight the displacements. Topography from the technical map of Lecce Province (1:5000).

### 4.8. Ciardo Coast

Unlike the other sites described above, the Ciardo coast is characterized by the outcropping substrate of Cretaceous limestone while its average height above the MSL ($H_c$) is about 3 m. Four of the five boulders identified here form a group of closely spaced boulders that lies above an onshore tilted by about 30°. TD values between 1.6 and 3.3 m were calculated for them (Tables A13 and A14). The direction of boulder displacement is from SW to NE (Figure A7). The fifth boulder, identified by satellite images through its socket, was likely carried out to sea as it was not found above the cliff.

### 4.9. Sant'Emiliano Coast

The average height above the MSL ($H_c$) of Sant'Emiliano cliff is about 2.5 m, and it is made up by Tertiary limestones. From the comparison of satellite images, two large boulders moved from their initial positions between July 2018 and June 2020 were identified (Tables A15 and A16). Both were not found above the cliff; thus, they have likely sunk below sea level. The geological observations of the ground confirmed the presence of the two relative sockets (cf. Figure A8b), on the basis of which the dimensions of the *a* and *b* axes were established (Figure A9).

### 4.10. Summary

The initial position of 64 out of 81 boulders was recognized. It is usually located on the rocky platforms and only for a few cases at the edge of the cliffs (boulders PRc, PRf, MAa, SUa, SUd, SUl, SUn). As disclosed in Section 4.1 and then detailed for each sites, for 59 boulders the TD was measured from the 28 June 2020 GE image. For 37 of the 59 boulders, the exact initial positioning was ascertained in the field by virtue of the geological features; thus, their TD were carefully determined (see Section 6). The remaining 22 out of 59 boulders form several clusters, and their initial positions were determined with an accuracy of a few meters (in Italics in Tables A1, A5, A7 and A13).

### 5. Marine Weather Conditions

Five storms were identified in the screening of the period between fall 2018 and spring 2020. They happened on: 29–30 October 2018, 12–13 November 2019, 24–26 November 2019, 22 December 2019, and 2 March 2020. The main meteorological characteristics will be briefly mentioned, and for each of them an estimation of the expected spectral maximum

wave height $H_0$ over the southern Apulia coast will be calculated by Equation (2), to assess their potential impact over the coastal boulders.

The 29–30 October 2018 storm was caused because of a 500 hPa trough generated over the Iberian peninsula approached the Mediterraneam region from north-west, thus, advecting very strong winds on the south-eastern part of the Italian peninsula. A deep surface low of abut 980 hPa crossed the north-western Mediterranean sea during the day 29 October. This storm, remembered as the Vaia storm, was characterized by very strong winds that affected the Ionian and Adriatic seas up to the Venezia Gulf. Studies analysed the particular meteorological conditions that locally caused very strong and persisting winds, that were responsible for the destruction of the so-called Violins Forest in the Veneto region [54].

The synoptic configuration remained unchanged for days, because of the association of high pressure conditions on eastern Europe and over the Atlantic sea, a condition that is becoming increasingly common in the last decades [55]. This caused strong winds with stable south-eastern direction and intensity to blow for over two days, with a maximum fetch from the Salento southern coast between 700 and 800 km, because of the stability in the direction (Figure A11). The wind speed was about 12–18 m/s at 10 m height offshore the Salento coast and above 20 m/s in the Venice lagoon.

The most effective for the impact over the southern Apulia coast was associated to a fetch between 700 and 800 km over the Ionian sea, that, with an average wind speed of about 15 m/s gives a maximum spectral wave height between 6.5 and 7.5 m by Equation (2). The stable south-eastern wind direction results in a potential wave impact over both the south-eastern and south-western Salento coasts. Vaia storm caused significant impact on coastal boulders in the northern Adriatic Sea [56], while only one boulder displacement was documented at the southern Apulia coast [52].

In 12–13 November 2019 a trough at 500 hpa approaching from the north west of France deepened strongly over the Gulf of Lion, that often acts as a feeding area in increasing the strength of the Mediterranean storm in this period of the year. The trough then migrated southward in the Tyrrenian sea reaching Sicily after acquiring more strength turning around the Atlante altiplan, almost assuming a Medicane-like structure, and thus advecting very strong winds over the Taranto gulf. A deep surface low of about 990 hPa was generated over the western Mediterranean during the day 12 of November.

The strong wind over Salento started soon after midnight from the south east, and persisted for about 24 h from almost the same direction with an offshore fetch of about 700–800 km, down to the Libya coast (Figure A12). In the following evening the wind turned from south-southwest until the afternoon of Wednesday 13 of November, with a fetch reduced to about 350 km, which kept reducing until the end of the event. The wind speed at 10 m height offshore was about 20 m/s. The most effective for the impact over the southern Salento coast was associated to a fetch of 700–800 km over the Ionian sea, that, with an average wind speed of 20 m/s gave a maximum spectral wave height of about 8–9 m.

These almost exceptional wind-wave conditions are in agreement and were confirmed by local observations in S. Maria di Leuca meteomarine station [10]. The wind direction was slowly changing during the first day from south-east to south, however, not affecting the overall fetch of over 700 km, also in agreement with its duration of over 24 h. The south-eastern wind direction associated to this fetch resulted in a potential wave impact over both the south-eastern and south-western Salento coasts.

A quite similar synoptic condition caused a new storm about only two weeks after, between 24 and 26 of November 2019, however, with a weaker depression in the middle atmosphere that migrated southward from the north-western Mediterranean region. Again the strongest winds blew from the south-eastern direction over the south Salento coast, but with a weaker intensity of about 15 m/s (Figure A13). In this case, the fetch was again of about 700–800 km, so that the wind-fetch conditions seem to be similar to the 2018 storm.

However, the duration of these conditions in the present case were of about 12 h, while in the 2018 storm, the long-fetch wind conditions lasted for more than 2 days, which makes a difference in terms of the wave height and effect over the coast. Indeed, taking

into account of the actual duration, the calculate spectral peak wave is reduced to 4–4.5 m (with an effective fetch of about 300 km).

Both the 22 December 2019 and the 2 March 2020 storms were caused by two middle-atmosphere lows coming from the north-eastern direction with respect to the Italian peninsula. This difference caused a different impact over the southern Apulia, with more variability of the wind speed and direction. In the case of December 2019, the prevailing direction was from the western direction over the Salento coast (Figure A14), with a shorter fetch because of the geographic shape of the Taranto gulf. The BOLAM model shows a wind of about 15 m/s at 10 m over the sea, associated to a fetch of about 300 km, which gives a maximum wave height of about 4–5 m. In this case, due to the prevailing western wind component, the waves are not expected to impact over the south-eastern Salento coast.

In the case of March 2020 the strongest wind blew from the South, with a lower intensity of about 12 m/s and a maximum fetch of nor more than 400 km (Figure A15). This fact potentially affected both sides of the southern Salento coast; however, the maximum calculated wave height is again not greater than 4–5 m at most.

## 6. Results and Discussion

### 6.1. Inferred Boulders Dynamics

Multi-temporal satellite image analysis allowed to oversee the main boulder displacements occurred in two years along 130 km of the Apulia coast. With reference to the two positions occupied by the boulders in July 2018 and June 2020, respectively, it is obviously not possible to establish whether the position changes were caused by one or more displacements. Only for the boulder SUi we know of two different movements, the first of which occurred due to the 29–30 October 2018 storm [52].

Again, the eighty-one boulders identified are not exhaustive of the whole phenomenon. It was possible to identify only those of large size, i.e., with major axis of at least 1 m. The latter correspond to those defined as "coarse boulder" by Blair and McPherson [41]. However, by virtue of a monitoring carried out in one of the sites (S. Isidoro, see Figure 1) we know that also numerous smaller clasts have changed position because of the 12–13 November storm [10].

As most of the boulders were located on the rocky platforms prior to transport, they must have been displaced by waves that, after the cliff overtopping, crossed the shore likely as bore flows [21,47,48,50]. By relating TD, FI and boulder size (Figure 10) some inferences may be done. As a whole, an inverse relationship between FI and TD is apparent, in good accordance with the experimental observations of Imamura et al. [57] and Nandasena and Tanaka [58].

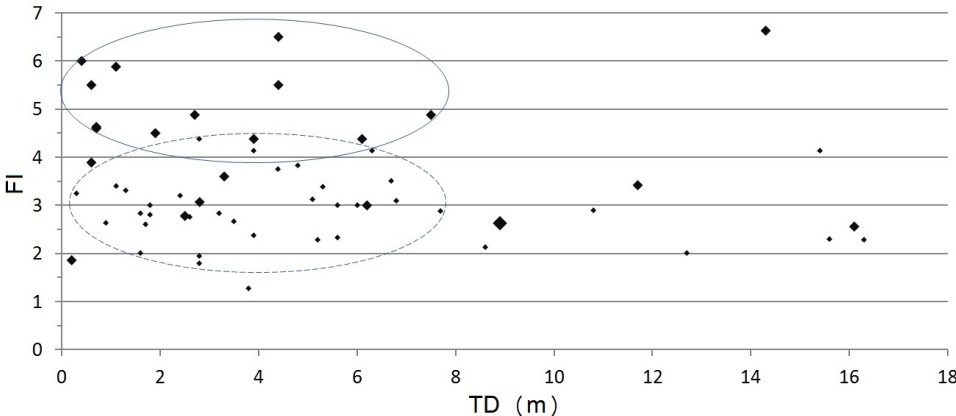

**Figure 10.** Flatness Index (FI) vs. Transport Distance (TD) diagram. Symbol size is related to the clast size (small = coarse boulder, medium = very coarse boulder, large = fine block). The solid blue line and the dashed blue line encloses the areas of highest incidence for very coarse and coarse boulders, respectively.

In particular, the "very coarse boulder" (according to the [41] classification, see the medium size symbol in Figure 10), appeared to be more easily displaced by the waves the more they were flattened. Calculated TDs were largely shorter than 10 m: less than 8 m for the 83% of the considered boulders; less than 4 m for the 54%. Some larger transports occurred. However, the ten of boulders with TD > 8 m apparently do not show relationships with their own shape and size. The inferred prevalent number of short transports for the flatter boulders is in agreement with the result of previous studies [56,59].

The diagram in Figure 11 relates the final distance ($x_f$) to the TD. It is apparent how $x_f$ overrates the change in position of the boulders, often even more than twice the actual transport. This suggests that using $x_f$ as a measure of boulder transport due to storm waves can lead to an overestimation of the actual dynamics. It would, therefore, be necessary to establish also the initial position $x_i$ of the boulders, in order to elaborate inferences relating to the boulder transport. In the present study, the combined use of satellite imagery and field geological surveys resulted in reliable TDs for 59 out of 81 boulders.

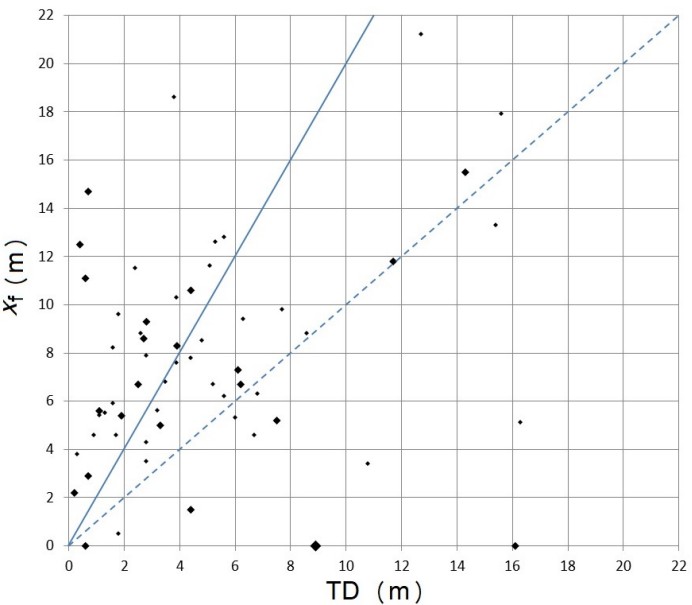

**Figure 11.** Final distance ($x_f$) vs. Transport Distance (TD) diagram. Symbol size is related to the clast size (small = coarse boulder, medium = very coarse boulder, and large = fine block).

Several identified displaced boulders are grouped, somewhere forming imbricated clusters. The analysis of the hydrodynamic and geological implications of the clusters [60,61] is beyond the aims of the present study. However, it should be noted that, for the case herein described, the clustering and even the imbrication of the boulders frequently occurred (see e.g., the boulders PRi,j,k,l and PRn,o,p at Punta Prosciutto; the boulders PIa,b,c,d,e at Punta Pizzo; the boulders MAf,g,h,i,j,k,l at the Mancaversa; the boulders CAa,b,c,d at the Capilungo; and CIa,b,c,d at the Ciardo).

### 6.2. Causative Storm Identification

From the point of view of the maximum wave height and energy the two storms of October 2018 and 12–13 November 2019 should be considered as the most effective to cause boulder displacements (Section 5). The strong surface winds and deep low pressure conditions were also responsible for additional relevant coastal surges that enhanced the wave effects over the coast. Taking into account the uncertainties in the parameters, above all the sea depth $D$ in the proximity of the coast, the estimated values for the sea level increase $Hs$ were between 1.2 and 1.7 m for the 2018 storm and between 1.5 and 2 m for the first 2019 storm (both of about one day of duration) from Equation (4). Equation (6) gives 1.5 m in 2018 and 1.0 m in 2019.

The last, although smaller, effect to be taken into account is the diurnal tide. This can be estimated as the product of the main periodic components that are functions of the latitude ([45], p. 335), giving a value of about 0.3 m, that is in agreement with typical local observations. In spite of the necessary approximations in the simplified models and the uncertainty in some parameters this analysis shows that both in 29–30 October 2018 and 12–13 November 2019 storms, the sea level elevation can be considered well above 1 m with respect to the mean sea level.

Following these approximate estimations, a value of $H_s$ = 1.5 m as been taken as representative of the average rising of the sea level in both the considered storms. These storms have certainly caused some boulder displacements at two sites as by post-event surveys [10,52,53] (see also Section 4).

### 6.3. Coastal Hazard Inferences

As proposed by several authors [7,10,22,23], the wave heights $H_m$ required to displace the boulders may be related to the coastal hazards. The values obtained from the equations proposed by [21,48] are listed in Tables A18–A25. They are between almost 10 m and about 0.5 m, depending on the boulder size and type of movement. About 20% of $H_m$ are between 5 and 2 m, which, considering the low altitudes above the sea level of the studied area, may be associated to significant flooding hazard. However, to achieve reliable conclusions, detailed expensive studies on each of the considered sites must be carried out (i.e., photogrammetric, laser scan, scuba surveys, and the processing of high-resolution images), together with an assessment of the reliability of the calculated minimum wave heights $H_m$ for the boulder movements.

Thus, in the present study, we attempted to assess the connection between $H_m$ and the wave height $H$ impacting the boulders, comparing the calculated minimum value $H_m$ with an estimate of the height of the onshore flow calculated taking into account the distance from the coastline of the boulder (Equation (A6)), and the actual marine weather conditions to estimate $H_0$ (by Equation (2)). Figure 12 compares the curves of the estimated wave height $H$ from Equation (A6) (Appendix D) with the requested minimum wave heights $H_m$ (circles), to move the boulders from the initial to the final position as a function of the onshore distance from the coastline. This approach is similar to that used by [48] to identify the causative wave events of some paleo-deposits.

The line type refers to the selected storm, and the colours refer to the coast average height for both the boulders and the storm wave height. The coasts of the study sites were clustered in three groups (black, red, and blue) for low (0–1 m), medium (1–2 m), and high coasts (2–3 m), respectively. In synthesis, the estimated wave height $H$ in a storm for a particular position and coastal height overcomes the minimum required wave height $H_m$ to move the boulder in the initial or final position when the circles identifying the boulder positions are below one of the curves of the same colour.

This figure shows that the condition $H > H_m$, holds for most of the boulders for the two strongest storms of the studied period, with a significant difference with respect to the other weaker storms, which appear to be unable to move most of the boulders even in the most favourable low coast conditions (as shown by the dash-dotted line that is representative of $H$ for $H_0$ = 4.5 m, and $H_s$ = 0.5 m).

Cox et al. (2020) [49] criticized the analytical equations, widely used to estimate the wave height required for boulder transport [2,7,21,48], that was also used in this work. They indicated several shortcomings, mainly regarding the 'constant' parameters used in the equations, such as the friction coefficients, the oversimplified assumption linking the wave height to the fluid velocity that is responsible for the stress force over the boulders, and the comparison of the calculated minimum wave height with the spectral peak wave height, as an estimation of the 'maximum' wave height of the storm.

As these authors showed in their work, the effect of the above assumptions were generally to overestimate the minimum wave height required to move the boulders in the case of meteorological storm with respect to the tsunami waves [49]. In addition, Engel

and May [48] showed that Equation (A6) tends to overestimate the actual decrease of $H_0$ at a distance X from the coastline. These facts, together with the underestimation of the maximum storm wave height $H_0$ due to its assimilation to the spectral peak wave, triggered many of the previous analyses in favour of the tsunami interpretation for the boulder displacements [48,49]. $H_0$ was also used here as a proxy for $R$, that is for the breaking wave height at the coastline (Equation (A6)).

Sunamura and Horikawa [62], proposed an estimation of the breaking wave height as an increasing function of the coastal slope and of the wave period that has been used by other authors [2]. In this case, taking into account the uncertainties in the determination of the wave parameters and, above all, the uncertainty about the significant coastal slope in the different sites, we avoided to introduce more indetermination so that a conservative approach with $H_0$ as a proxy for the breaking wave height $R$ was used, that is actually its minimum value for a vanishing slope.

Even with this minimum value for the wave height, the result of the analysis showed that, for most of the considered boulders, the storm-produced displacement was indeed possible, in agreement with the calculations, and the application of the certainly well-founded aforementioned observations and criticisms already discussed, as also the use of a Rayleigh distribution for the sea wave heights with a non-negligible probability of waves higher than the spectral peak wave $H_0$, would generally result in higher values for the waves impacting over the boulders and, consequently, a reinforcement of this result, with possibly even more points with $H > H_m$ in Figure 12.

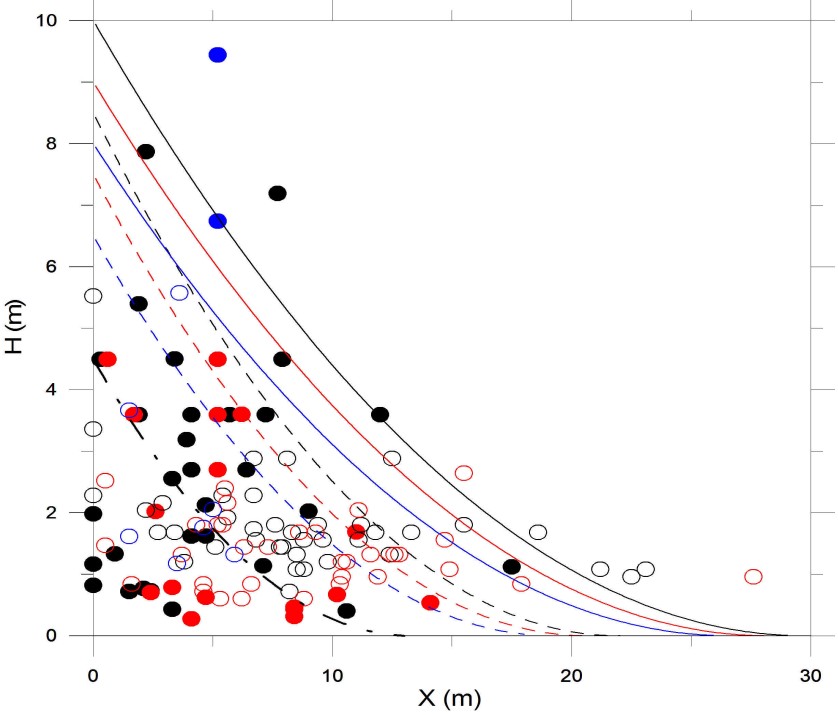

**Figure 12.** Comparison between the calculated onshore wave height $H$ (lines) and the minimum wave height required to move the boulders $H_m$ (circles) as a function of the distance X from the coastline and the coast height $H_c$ (colours) for different storm spectral peak waves $H_0$. Continuous line: Wave height as a function of the onshore distance calculated by Equation (A6), with T estimated by Equation (1), for the 12–13 November 2019 storm ($H_0$ = 9.0 m, $H_s$ = 1.5 m). Black: $H_c$ = 0.5 m, Red: $H_c$ = 1.5 m, and Blue: $H_c$ = 2.5 m. Dashed line: same as the continuous line but refers to the October 2018 storm ($H_0$ = 7.5 m, $H_s$ = 1.5 m). Dot-dashed line: Wave height as a function of the onshore distance calculated by Equation (A6) for $H_0$ = 4.5 m and $H_c$ = 0.5 m. Filled circles: $H_m$ calculated by Equations (A1)–(A4) as a function of the initial onshore distance X of the boulder and the coast height (colours as above). Empty circles: $H_m$ calculated by Equation (A5) as a function of the final onshore distance X of the boulder and the coast height (colours as above).

## 7. Conclusions

Three main conclusions can be inferred with reference to the main aims of this work (see Section 1):

(a) The carried out investigations resulted in eighty-one documented boulder displacements along the about 130 km studied coast length. Publicly available satellite imagery coupled with field surveys proved useful for this purpose. For three quarters of the studied boulders, the TDs were calculated with varying degrees of accuracy, while, only for a small number of them, unfavourable lighting or not optimal resolution of the satellite images limited the investigations. An inverse relationship between the displacement magnitude and flattened shape was found, which was in good agreement with the literature knowledge.

(b) The analysis of synoptic weather maps allowed the identification of the stormy days in the period of study. They were associated to analytical equations to estimate the spectral peak of the SWH, thus, identifying two main storms of relevant wave height. The final results in Figure 12 confirmed that, only in the case of the two strongest storms of October 2018 and 12–13 November 2019, the estimated wave height was above the minimum required to move most of the boulders. The effect was also enhanced by a relevant sea level rise by storm surge and was much more effective than in the remaining weaker storms, when all the displaced boulders are taken into account. The intensity of the storm is important, but must be also associated to a proper wind direction (fetch) and its persistence.

(c) In the sites of study the calculated minimum wave heights for the boulder movements are actually found to be generally lower than the estimated onshore wave heights of the strongest storms. This confirms the possibility of using the boulders as a proxy for evaluating the storm minimum wave heights impacting at different onshore distances from the coastline and suggests a significant flooding hazard.

The present analysis also showed that, in the considered locations, most of the boulders can be displaced by the strongest storms, however, with a quite large gap between their effect and that of weaker storms in the observed period. The storm of 12–13 November 2019 was indeed of non-common strength for the local wind speed over the sea and the wave heights, but it also appears that even somehow lower wind speeds, such as in the October 2018 storm, can cause quite high waves, possibly effective for the boulder motion, when the duration and persistence of the wind direction produces a long fetch in the site, as in the case of meteorological blocking effects, and a relevant storm surge can also be present.

As indicated in the Introduction (Section 1), some general climatic considerations and early climate modelling studies suggested a possible general increase in the power of the storms in the next future, in spite of a possible decrease in their total number [24,25]. Studies have also been made for the specific case of Mediterranean storms, where some similar signals were noted from the first numerical modelling approaches, although they did not appear as statistically significant [63]. More recent studies, however, seem to confirm a possible increasing trend in the power of the storms in the Mediterranean sea, together with a decrease in their frequency [64,65].

In our study, the displacements of quite large boulders appear to be connected with energetic waves that are characteristic of single very energetic storms. Thus, this study suggests that the apparent general climatic trend, if confirmed, in connection with the sea-level rise effect expected by both strong storm surges and global warming, implies that the effect of the meteorological storms in coastal hydrodynamic in the considered site could even be enhanced in the near future.

**Author Contributions:** Conceptualization, methodology, M.D.R., P.M. and L.O.; satellite imagery investigation, M.D.R.; geological investigation, M.D.R. and L.O.; marine weather analysis, P.M.; hydrodynamics calculation, M.D.R. and P.M.; writing—original draft preparation, M.D.R., P.M. and L.O.; writing—review and editing, M.D.R. and P.M. All authors have read and agreed to the published version of the manuscript.

**Funding:** This research received no external funding.

**Acknowledgments:** The authors wish to acknowledge A.L. Signore and L. Marzo for the assistance in the field surveys.

**Conflicts of Interest:** The authors declare no conflict of interest.

### Appendix A

The data of each boulder (Sections 3.1 and 4), together with selected features of the more typical ones, are reported in this appendix. Geographical coordinates and TD were taken from the 28 June 2020 GE image. Where the initial position was defined for groups of boulders (PRi,j,k,l and PRq,r of Figures 3 and 4; PIa,b,c,d,e of Figures 5 and A2; MAf,g,h,i,j,k,l of Figures 6 and A3b; CIa,b,c,d of Figure A7) italics were used (Tables A1, A2, A5–A8, A13 and A14).

For some boulders, the MT is uncertain between ST and OV (Tables A2 and A8). In some photos (Figures A2b,c, A3a,b,c, and A5–A7), 1 m-long tape measure for scale was placed. As regards the Punta Prosciutto headland, two groups of boulders are shown in Figure 4. Before the transport, the imbricated PRi,j,k,l occupied a rectangular surface (3 × 4 m approximately) SW of the present position (Figure 3). The resolution of the 20 July 2018 image does not allow to individually identified these boulders. The same for PRq,r.

**Table A1.** Initial and final geographical coordinates of the displaced boulders at Punta Prosciutto headland; ind., indeterminable.

| Boulder ID | Initial Position Latitude | Longitude | Final Position Latitude | Longitude | TD [m] |
|---|---|---|---|---|---|
| PRa | 40°17′32.24″ N | 17°46′00.28″ E | 40°17′32.49″ N | 17°46′00.37″ E | 7.7 |
| PRb | 40°17′32.29″ N | 17°46′00.31″ E | 40°17′32.38″ N | 17°46′00.34″ E | 2.8 |
| PRc | 40°17′30.14″ N | 17°45′48.31″ E | 40°17′30.25″ N | 17°45′47.87″ E | 10.8 |
| PRd | 40°17′30.40″ N | 17°45′46.87″ E | 40°17′30.52″ N | 17°45′46.78″ E | 3.5 |
| PRe | 40°17′32.47″ N | 17°45′46.60″ E | 40°17′32.60″ N | 17°45′46.63″ E | 3.8 |
| PRf | 40°17′33.00″ N | 17°45′45.86″ E | 40°17′33.51″ N | 17°45′46.13″ E | 16.3 |
| PRg | ind. | ind. | 40°17′33.87″ N | 17°45′46.33″ E | - |
| PRh | ind. | ind. | 40°17′34.28″ N | 17°45′46.09″ E | - |
| PRi | *40°17′34.72″ N* | *17°45′45.62″ E* | 40°17′34.81″ N | 17°45′45.71″ E | 3.9 |
| PRj | *40°17′34.72″ N* | *17°45′45.62″ E* | 40°17′34.83″ N | 17°45′45.73″ E | 4.8 |
| PRk | *40°17′34.72″ N* | *17°45′45.62″ E* | 40°17′34.87″ N | 17°45′45.75″ E | 6.3 |
| PRl | *40°17′34.72″ N* | *17°45′45.62″ E* | 40°17′34.85″ N | 17°45′45.69″ E | 4.4 |
| PRm | 40°17′35.13″ N | 17°45′45.60″ E | 40°17′35.19″ N | 17°45′45.69″ E | 2.6 |
| PRn | ind. | ind. | 40°17′35.39″ N | 17°45′45.64″ E | - |
| PRo | ind. | ind. | 40°17′35.41″ N | 17°45′45.74″ E | - |
| PRp | ind. | ind. | 40°17′35.35″ N | 17°45′45.78″ E | - |
| PRq | *40°17′35.48″ N* | *17°45′45.38″ E* | 40°17′35.77″ N | 17°45′45.69″ E | 11.7 |
| PRr | *40°17′35.48″ N* | *17°45′45.38″ E* | 40°17′35.84″ N | 17°45′45.84″ E | 15.4 |

**Table A2.** Main features of the displaced boulders at Punta Prosciutto headland; Dimensions of the boulder axes $a,b,c$, initial distances from the cliff edge $x_i$, final distances from the cliff edge $x_f$. Li, Lithology; C, Calcarenite; L, Limestone; S, Sandstone; Sh, Shape; B, Bladed; P, Prolate, D, Disk; E, Equant; FI, Flatness Index; PTS, Pre Transport Setting; JB, Joint-Bounded, SA, Sub-Aerial; SM, Submerged; MT, Movement Type; ST, saltation; SL, sliding; OV, overturning; and ind., indeterminable.

| Boulder ID | $a$ [m] | $b$ [m] | $c$ [m] | $x_i$ [m] | $x_f$ [m] | Li | Sh | FI | PTS | MT |
|---|---|---|---|---|---|---|---|---|---|---|
| PRa | 1.3 | 1.0 | 0.4 | 7.2 | 9.8 | C | D | 2.88 | SA | ST |
| PRb | 1.9 | 1.2 | 0.8 | 7.7 | 7.9 | C | P | 1.94 | SA | ST |
| PRc | 1.5 | 1.4 | 0.5 | 0.3 | 3.4 | C | D | 2.90 | SA | ST,OV |

**Table A2.** *Cont.*

| Boulder ID | $a$ [m] | $b$ [m] | $c$ [m] | $x_i$ [m] | $x_f$ [m] | Li | Sh | FI | PTS | MT |
|---|---|---|---|---|---|---|---|---|---|---|
| PRd | 1.9 | 1.3 | 0.6 | 4.7 | 6.8 | C | D | 2.67 | SA | OV |
| PRe | 1.9 | 1.4 | 1.3 | 17.5 | 18.6 | C | E | 1.27 | SA | OV |
| PRf | 2.0 | 1.2 | 0.7 | 0.9 | 5.1 | C | B | 2.29 | SA | OV |
| PRg | 1.8 | 1.1 | 0.4 | ind. | 12.4 | C | B | 3.63 | SA | ind. |
| PRh | 1.7 | 1.3 | 0.5 | ind. | 8.8 | C | D | 3.00 | SA | ind. |
| PRi | 1.8 | 1.5 | 0.4 | *4.1* | 7.6 | C | D | 4.13 | SA | ST |
| PRj | 1.4 | 0.9 | 0.3 | *4.1* | 8.5 | C | B | 3.83 | SA | ST |
| PRk | 1.8 | 1.5 | 0.4 | *4.1* | 9.4 | C | D | 4.13 | SA | ST,OV |
| PRl | 1.8 | 1.2 | 0.4 | *4.1* | 7.8 | C | D | 3.75 | SA | OV |
| PRm | 1.3 | 0.9 | 0.4 | 7.1 | 8.8 | C | D | 2.75 | SA | OV |
| PRn | 3.1 | 2.4 | 0.5 | ind. | 8.1 | C | D | 5.50 | SA | ST,OV |
| PRo | 1.9 | 1.3 | 0.4 | ind. | 11.1 | C | D | 4.00 | SA | ST,OV |
| PRp | 2.2 | 1.5 | 0.6 | ind. | 11.2 | C | D | 3.08 | SA | ST,OV |
| PRq | 2.7 | 1.4 | 0.6 | *1.9* | 11.8 | C | B | 3.42 | SA | ST |
| PRr | 1.9 | 1.4 | 0.4 | *1.9* | 13.3 | C | D | 4.13 | SA | ST |

As mentioned above (Section 4), the boulder dynamics occurred at the stretch of the Torre Sant'Isidoro coast because of the November 2019 storm has already been described by [10]. Here, we observe how even displaced coarse boulders may not be recognized by the multi-temporal satellite imagery (Figure A1). This can be due to the not optimal resolution of the images or to an unfavourable lighting.

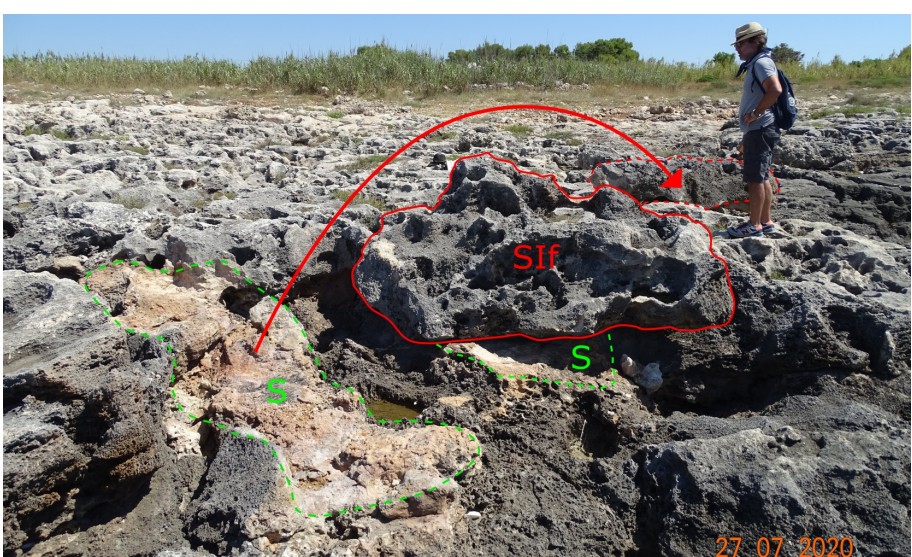

**Figure A1.** Torre Sant'Isidoro coast. Below the boulder SIf, its socket was recognized during the field survey (from [10], modified). By the June 2020 satellite image, the other boulder (highlighted in red dashed line) was not identified despite its socket (very evident in the photo). Note that the field survey was made one month after the date of the last GE image.

**Table A3.** Initial and final geographical coordinates of the displaced boulders at the Torre Sant'Isidoro coast; ind., indeterminable.

| Boulder ID | Initial Position Latitude | Longitude | Final Position Latitude | Longitude | TD [m] |
|---|---|---|---|---|---|
| SIc | 40°13′34.20″ N | 17°55′17.51″ E | ind. | ind. | - |
| SIf | 40°13′37.03″ N | 17°55′14.06″ E | 40°13′37.04″ N | 17°55′14.07″ E | 0.4 |
| SIg | 40°13′43.82″ N | 17°55′05.10″ E | 40°13′43.89″ N | 17°55′05.12″ E | 2.4 |

**Table A4.** Main features of the displaced boulders at the Torre Sant'Isidoro coast; see the caption of Table A2 for symbols.

| Boulder ID | $a$ [m] | $b$ [m] | $c$ [m] | $x_i$ [m] | $x_f$ [m] | Li | Sh | FI | PTS | MT |
|:---:|:---:|:---:|:---:|:---:|:---:|:---:|:---:|:---:|:---:|:---:|
| SIc | 2.6 | 1.7 | 0.6 | 1.5 | ind. | C | B | 3.58 | SA | SL |
| SIf | 2.8 | 2.4 | 0.4 | 12 | 12.5 | C | D | 6.00 | SA | ST |
| SIg | 1.7 | 1.5 | 0.5 | 9 | 11.5 | C | D | 3.20 | SA | OV |

Among the several displaced boulders identified at the Punta Pizzo headland, photos of a very coarse overturned boulder (Figures A2a,b) and of the main groups (Figures A2c,d) were selected. Note that the socket of PIr is clearly visible, while the print of the initial position of the boulder group is less so. However, the latter is well recognizable by the 20 July 2018 satellite image (Figure 5), where it shows a rounded contour with a radius of about 2 m.

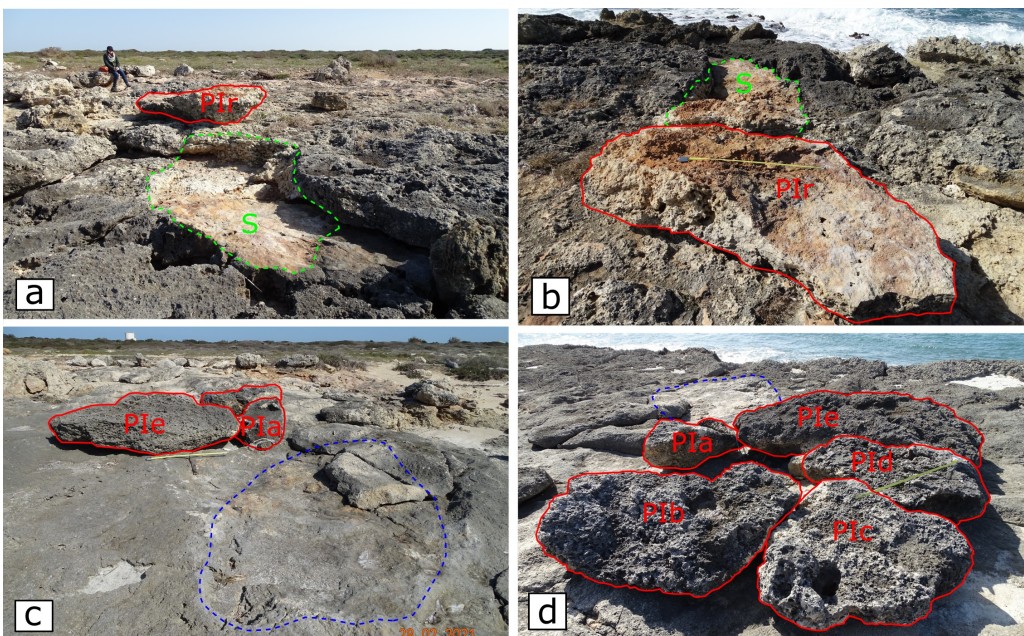

**Figure A2.** Punta Pizzo headland: (**a**) The boulder PIr and its socket taken from the coastline. (**b**) The same objects of a, taken from inland. (**c**) The boulder group PIa,b,c,d,e and the initial position (blue dashed line) taken from the coastline. (**d**) The same objects of c, taken from inland.

**Table A5.** Initial and final geographical coordinates of the displaced boulders at the Punta Pizzo headland; ind., indeterminable.

| Boulder ID | Initial Position | | Final Position | | TD [m] |
|:---:|:---:|:---:|:---:|:---:|:---:|
| | Latitude | Longitude | Latitude | Longitude | |
| PIa | 39°59′34.46″ N | 17°59′48.06″ E | 39°59′34.58″ N | 17°59′48.10″ E | 3.9 |
| PIb | 39°59′34.46″ N | 17°59′48.06″ E | 39°59′34.61″ N | 17°59′48.18″ E | 5.3 |
| PIc | 39°59′34.46″ N | 17°59′48.06″ E | 39°59′34.63″ N | 17°59′48.14″ E | 5.6 |
| PId | 39°59′34.46″ N | 17°59′48.06″ E | 39°59′34.62″ N | 17°59′48.10″ E | 5.1 |
| PIe | 39°59′34.46″ N | 17°59′48.06″ E | 39°59′34.60″ N | 17°59′48.06″ E | 4.4 |
| PIf | ind. | ind. | 39°59′35.19″ N | 17°59′47.30″ E | - |
| PIg | 39°59′43.62″ N | 17°59′41.76″ E | 39°59′43.68″ N | 17°59′41.67″ E | 2.8 |
| PIh | 39°59′43.50″ N | 17°59′41.37″ E | 39°59′43.77″ N | 17°59′41.86″ E | 14.3 |
| PIi | ind. | ind. | 39°59′43.83″ N | 17°59′41.79″ E | - |
| PIj | 39°59′43.55″ N | 17°59′41.43″ E | 39°59′43.92″ N | 17°59′41.88″ E | 15.6 |

**Table A5.** *Cont.*

| Boulder ID | Initial Position Latitude | Longitude | Final Position Latitude | Longitude | TD [m] |
|---|---|---|---|---|---|
| PIk | ind. | ind. | 39°59′43.97″ N | 17°59′42.30″ E | - |
| PIl | ind. | ind. | 39°59′44.25″ N | 17°59′41.05″ E | - |
| PIm | 39°59′44.70″ N | 17°59′41.28″ E | 39°59′44.70″ N | 17°59′41.30″ E | 0.6 |
| PIn | 39°59′44.86″ N | 17°59′41.39″ E | 39°59′44.86″ N | 17°59′41.41″ E | 0.7 |
| PIo | 39°59′53.26″ N | 17°59′38.24″ E | 39°59′53.29″ N | 17°59′38.25″ E | 0.9 |
| PIp | 39°59′53.25″ N | 17°59′38.15″ E | 39°59′53.33″ N | 17°59′38.21″ E | 2.8 |
| PIq | ind. | ind. | 39°59′53.46″ N | 17°59′38.23″ E | - |
| PIr | 39°59′53.61″ N | 17°59′38.20″ E | 39°59′53.64″ N | 17°59′38.30″ E | 2.7 |

**Table A6.** Main features of the displaced boulders at the Punta Pizzo headland; see the caption of Table A2 for symbols.

| Boulder ID | $a$ [m] | $b$ [m] | $c$ [m] | $x_i$ [m] | $x_f$ [m] | Li | Sh | FI | PTS | MT |
|---|---|---|---|---|---|---|---|---|---|---|
| PIa | 1.2 | 0.7 | 0.4 | *8.4* | 10.3 | C | B | 2.38 | SA | SL |
| PIb | 1.6 | 1.1 | 0.4 | *8.4* | 12.6 | C | D | 3.38 | SA | SL |
| PIc | 1.3 | 1.1 | 0.4 | *8.4* | 12.8 | C | D | 3.00 | SA | SL |
| PId | 1.4 | 1.1 | 0.4 | *8.4* | 11.6 | C | D | 3.13 | SA | SL |
| PIe | 2.7 | 1.0 | 0.4 | *8.4* | 10.6 | C | D | 6.50 | SA | SL |
| PIf | 2.1 | 1.0 | 0.7 | ind. | 10.4 | C | P | 2.11 | SA | ST |
| PIg | 2.9 | 1.4 | 0.7 | 11.0 | 9.3 | C | B | 3.07 | SA | OV |
| PIh | 3.1 | 2.2 | 0.4 | 1.7 | 15.5 | C | D | 6.63 | SA | ST |
| PIi | 1.3 | 0.9 | 0.4 | ind. | 14.9 | C | D | 2.75 | SA | ind. |
| PIj | 1.6 | 0.7 | 0.5 | 3.2 | 17.9 | C | P | 2.30 | SA | ind. |
| PIk | 1.1 | 0.8 | 0.6 | ind. | 27.6 | C | E | 1.58 | SA | OV |
| PIl | 1.6 | 0.7 | 0.3 | ind. | 1.6 | C | B | 3.83 | SM | ind. |
| PIm | 2.7 | 1.7 | 0.4 | 10.2 | 11.1 | C | B | 5.50 | SA | SL |
| PIn | 2.4 | 1.3 | 0.4 | 14.1 | 14.7 | C | B | 4.63 | SA | SL |
| PIo | 1.5 | 0.6 | 0.4 | 4.1 | 4.6 | C | P | 2.63 | SA | SL |
| PIp | 2.0 | 1.5 | 0.4 | 2.6 | 4.3 | C | D | 4.38 | SA | OV |
| PIq | 1.3 | 0.7 | 0.3 | ind. | 6.6 | C | B | 3.33 | SA | ind. |
| PIr | 2.5 | 1.4 | 0.4 | 6.2 | 8.6 | C | B | 4.88 | JB | OV |

The largest boulder (Figure A3a), the main cluster (Figure A3b), and a rare sandstone boulder (Figure A3c) are shown for Mancaversa. Regarding the initial position of the MAf,g,h,i,j,k,l group, by the July 20 2018 satellite image, the area approximately occupied before the transport was recognized (Figure 6).

**Table A7.** Initial and final geographical coordinates of the displaced boulders at the Mancaversa coast; ind., indeterminable.

| Boulder ID | Initial Position Latitude | Longitude | Final Position Latitude | Longitude | TD [m] |
|---|---|---|---|---|---|
| MAa | 39°58′20.49″ N | 18°0′43.70″ E | 39°58′20.64″ N | 18°00′43.72″ E | 4.4 |
| MAb | 39°58′25.26″ N | 18°0′41.07″ E | 39°58′25.29″ N | 18°00′41.08″ E | 1.1 |
| MAc | 39°58′26.55″ N | 18°0′38.73″ E | 39°58′26.61″ N | 18°00′38.73″ E | 1.8 |
| MAd | ind. | ind. | 39°58′33.36″ N | 18°00′33.24″ E | - |
| MAe | ind. | ind. | 39°58′33.38″ N | 18°00′33.10″ E | - |
| MAf | *39°58′33.93″ N* | *18°00′32.40″ E* | 39°58′34.15″ N | 18°00′32.63″ E | 8.6 |
| MAg | *39°58′33.93″ N* | *18°00′32.40″ E* | 39°58′34.09″ N | 18°00′32.59″ E | 6.7 |
| MAh | *39°58′33.93″ N* | *18°00′32.40″ E* | 39°58′34.10″ N | 18°00′32.54″ E | 6.0 |
| MAi | *39°58′33.93″ N* | *18°00′32.40″ E* | 39°58′34.09″ N | 18°00′32.52″ E | 5.6 |
| MAj | *39°58′33.93″ N* | *18°00′32.40″ E* | 39°58′34.15″ N | 18°00′32.55″ E | 7.5 |

**Table A7.** *Cont.*

| Boulder ID | Initial Position Latitude | Longitude | Final Position Latitude | Longitude | TD [m] |
|---|---|---|---|---|---|
| MAk | 39°58′33.93″ N | 18°00′32.40″ E | 39°58′34.14″ N | 18°00′32.50″ E | 6.8 |
| MAl | 39°58′33.93″ N | 18°00′32.40″ E | 39°58′34.12″ N | 18°00′32.48″ E | 6.1 |
| MAm | 39°59′07.40″ N | 18°00′08.71″ E | 39°59′07.43″ N | 18°00′08.73″ E | 1.1 |
| MAn | 39°59′07.40″ N | 18°00′08.71″ E | 39°59′07.43″ N | 18°00′08.74″ E | 1.3 |
| MAo | ind. | ind. | 39°59′07.57″ N | 18°00′08.73″ E | - |

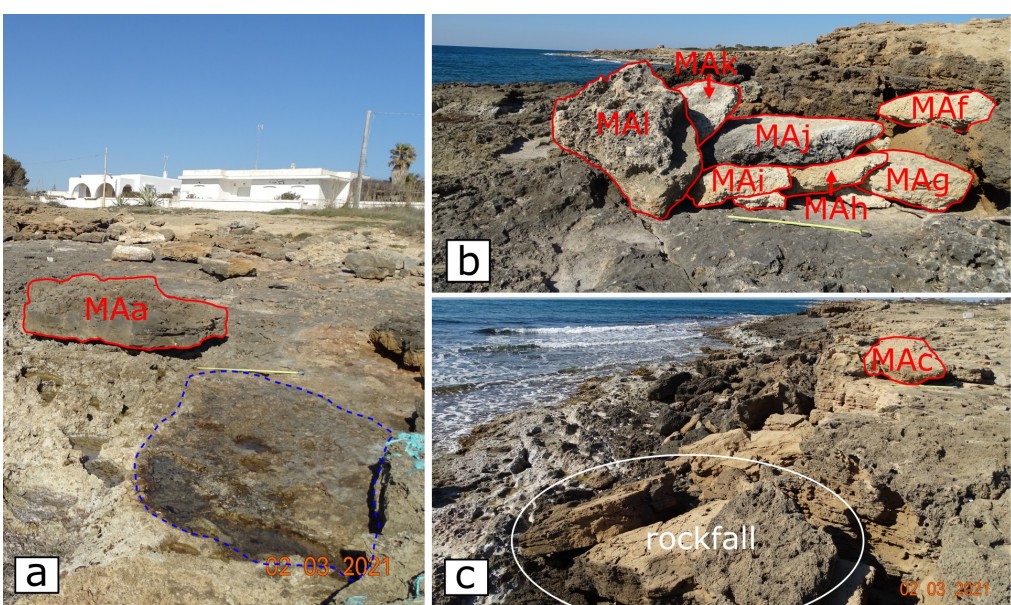

**Figure A3.** Mancaversa stretch of coast: (**a**) The boulder MAa and its initial position (blue dashed line) taken from the coastline. (**b**) The seven boulder group MAf-l; note that the unaltered rocky surfaces suggest the overturning of the boulders. (**c**) The sandstone boulder MAc and the neighbour, likely contemporary, rockfall.

**Table A8.** Main features of the displaced boulders at the Mancaversa coast; see the caption of Table A2 for symbols.

| Boulder ID | $a$ [m] | $b$ [m] | $c$ [m] | $x_i$ [m] | $x_f$ [m] | Li | Sh | FI | PTS | MT |
|---|---|---|---|---|---|---|---|---|---|---|
| MAa | 3.4 | 2.1 | 0.5 | 0.6 | 1.5 | C | B | 5.50 | JB | SL |
| MAb | 1.9 | 1.5 | 0.5 | 4.7 | 5.4 | C | D | 3.40 | SA | SL |
| MAc | 1.3 | 1.1 | 0.4 | 2.1 | 4.1 | S | D | 3.00 | SA | OV |
| MAd | 0.9 | 0.8 | 0.4 | ind. | 11.9 | C | D | 2.13 | SA | ind. |
| MAe | 2.3 | 0.8 | 0.5 | ind. | 10.4 | C | B | 3.10 | SA | ind. |
| MAf | 1.2 | 0.5 | 0.4 | 5.2 | 8.8 | C | P | 2.13 | SA | ST,OV |
| MAg | 1.4 | 0.7 | 0.3 | 5.2 | 4.6 | C | B | 3.50 | SA | ST |
| MAh | 1.3 | 0.5 | 0.3 | 5.2 | 5.3 | C | B | 3.00 | SA | ST,OV |
| MAi | 0.9 | 0.5 | 0.3 | 5.2 | 6.2 | C | B | 2.33 | SA | ST,OV |
| MAj | 2.4 | 1.5 | 0.4 | 5.2 | 5.2 | C | B | 4.88 | SA | ST,OV |
| MAk | 1.9 | 1.2 | 0.5 | 5.2 | 6.3 | C | B | 3.10 | SA | ST,OV |
| MAl | 2.3 | 1.2 | 0.4 | 5.2 | 7.3 | C | B | 4.38 | SA | ST,OV |
| MAm | 2.9 | 1.8 | 0.4 | 2.4 | 5.6 | C | B | 5.88 | SA | SL |
| MAn | 1.8 | 1.5 | 0.5 | 3.3 | 5.5 | S | D | 3.30 | SA | SL |
| MAo | 1.7 | 1.1 | 0.5 | ind. | 3.7 | C | B | 2.80 | SA | OV |

The Figure 8a,b show the fine block SUa and the very coarse boulder SUl (according to [41] classification) and their socket.

The comparison between satellite images was decisive for the recognition of the SUa displacement (Figure A4).

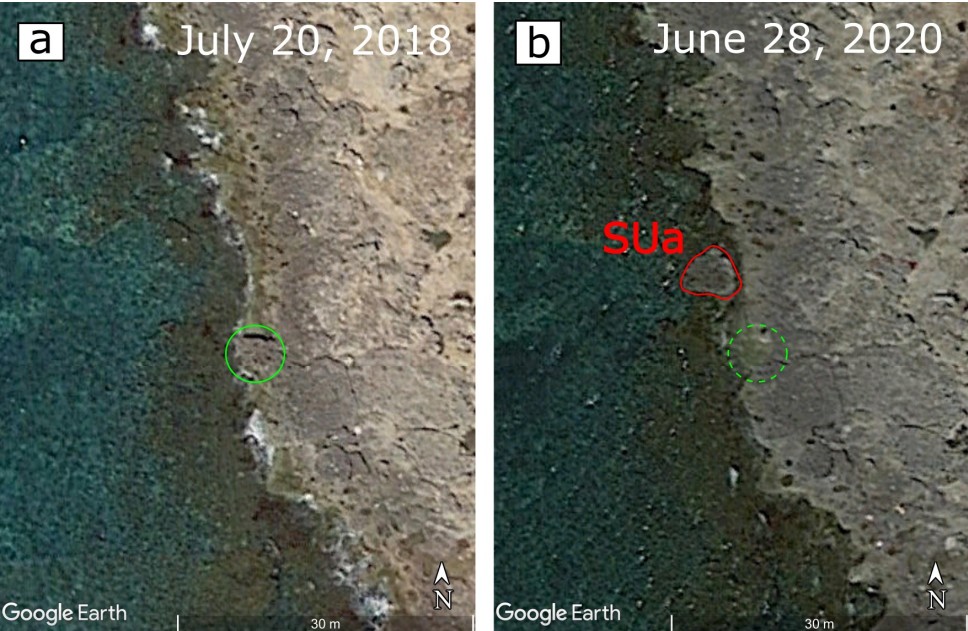

**Figure A4.** Comparative satellite images of the boulder SUa (Torre Suda): (**a**) The July 2018 image (eye elevation of 40 m); the green circle highlights the initial position. (**b**) The June 2020 image (eye elevation of 40 m).

As mentioned in Section 4, the boulder SUi was displaced for a first time during the October 2018 storm. In Figure A5, SUi is shown after this storm (cf. Figure 7). To underline such a singular characteristic, its initial position and the distance from the cliff are in bold in Tables A9 and A10 .

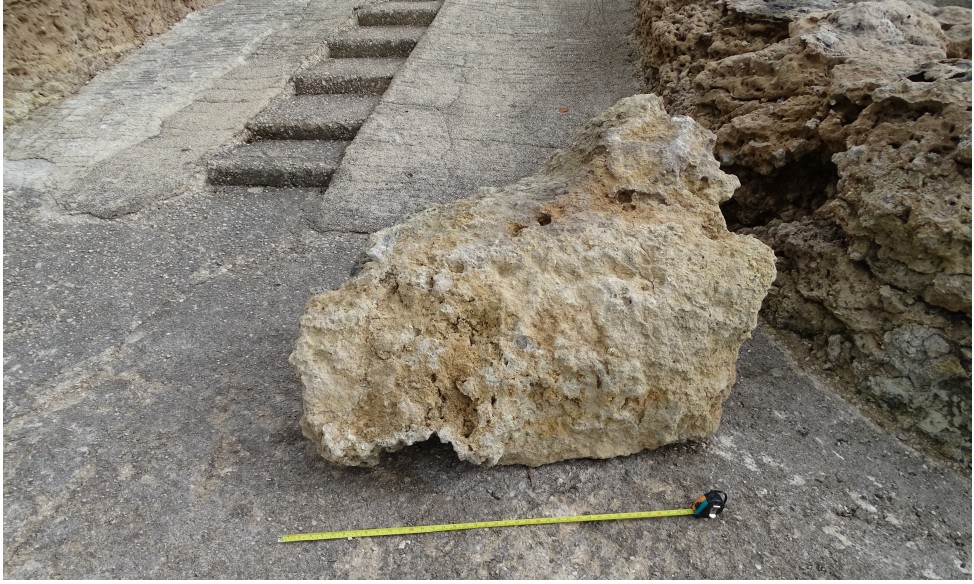

**Figure A5.** The boulder SUi, onshore displaced by the October 2018 storm. It was later re-mobilized as confirmed by the June 2020 GE image, where the boulder is placed further inland.

**Table A9.** Initial and final geographical coordinates of the displaced boulders at the Torre Suda coast; ind., indeterminable.

| Boulder ID | Initial Position Latitude | Longitude | Final Position Latitude | Longitude | TD [m] |
|---|---|---|---|---|---|
| SUa | 39°57′06.44″ N | 18°01′46.14″ E | 39°57′06.71″ N | 18°01′46.02″ E | 8.9 |
| SUb | ind. | ind. | 39°57′08.03″ N | 18°01′46.61″ E | - |
| SUc | ind. | ind. | 39°57′08.08″ N | 18°01′46.66″ E | - |
| SUd | 39°57′35.42″ N | 18°01′37.61″ E | ind. | ind. | - |
| SUe | 39°57′36.70″ N | 18°01′37.65″ E | 39°57′36.72″ N | 18°01′37.64″ E | 0.7 |
| SUf | 39°57′36.56″ N | 18°01′37.82″ E | 39°57′36.76″ N | 18°01′37.89″ E | 6.2 |
| SUg | ind. | ind. | 39°57′36.81″ N | 18°01′37.86″ E | - |
| SUh | 39°57′38.82″ N | 18°01′37.32″ E | 39°57′38.95″ N | 18°01′37.47″ E | 5.2 |
| SUi | **39°57′40.27″ N** | **18°01′36.40″ E** | 39°57′40.60″ N | 18°01′36.71″ E | 12.7 |
| SUj | 39°57′40.84″ N | 18°01′36.25″ E | 39°57′40.85″ N | 18°01′36.26″ E | 0.3 |
| SUk | 39°57′40.87″ N | 18°01′36.17″ E | 39°57′40.87″ N | 18°01′36.18″ E | 0.2 |
| SUl | 39°57′40.87″ N | 18°01′36.05″ E | 39°57′40.89″ N | 18°01′36.03″ E | 0.6 |
| SUm | ind. | ind. | 39°57′41.14″ N | 18°01′36.06″ E | - |
| SUn | 39°57′40.74″ N | 18°01′35.90″ E | 39°57′41.26″ N | 18°01′35.79″ E | 16.1 |

**Table A10.** Main features of the displaced boulders at the Torre Suda coast; see the caption of Table A2 for symbols.

| Boulder ID | $a$ [m] | $b$ [m] | $c$ [m] | $x_i$ [m] | $x_f$ [m] | Li | Sh | FI | PTS | MT |
|---|---|---|---|---|---|---|---|---|---|---|
| SUa | 5.4 | 4.6 | 1.9 | 0.0 | 0.0 | C | D | 2.63 | SM | SL |
| SUb | 1.8 | 0.8 | 0.5 | ind. | 22.5 | C | B | 2.60 | SA | ST |
| SUc | 1.7 | 0.9 | 0.7 | ind. | 23.1 | C | P | 1.86 | SA | ST |
| SUd | 4.9 | 3.2 | 1.8 | 0.0 | ind. | C | B | 2.25 | SM | ind. |
| SUe | 2.8 | 1.8 | 0.5 | 2.4 | 2.9 | C | B | 4.60 | SA | SL |
| SUf | 2.3 | 1.9 | 0.7 | 3.4 | 6.7 | C | D | 3.00 | SA | ST |
| SUg | 1.4 | 1.1 | 0.5 | ind. | 8.5 | C | D | 2.50 | SM | ST |
| SUh | 1.9 | 1.3 | 0.7 | 2.2 | 6.7 | S | D | 2.29 | SA | ST |
| SUi | 1.1 | 0.9 | 0.5 | **10.6** | 21.2 | C | D | 2.00 | SA | SL |
| SUj | 1.6 | 1.0 | 0.4 | 3.3 | 3.8 | C | B | 3.25 | SA | SL |
| SUk | 2.4 | 1.7 | 1.1 | 2.1 | 2.2 | C | D | 1.86 | SA | SL |
| SUl | 3.8 | 2.8 | 0.9 | 0.0 | 0.0 | C | D | 3.89 | SA | SL |
| SUm | 1.6 | 1.4 | 0.6 | ind. | 2.7 | C | D | 2.50 | SA | ST |
| SUn | 2.2 | 1.9 | 0.8 | 0.0 | 0.0 | C | D | 2.56 | SM | SL |

As far as it concerns the CAa,b,c, group and boulder CAd (Capilungo coast, Figure 9), the field evidence suggests that the contour of the whole surface was occupied before the displacement (Figure A6). The 20 July 2018 GE image, instead, allows the recognition of each initial position (Tables A11 and A12).

**Table A11.** Initial and final geographical coordinates of the displaced boulders at the Capilungo coast; ind., indeterminable.

| Boulder ID | Initial Position Latitude | Longitude | Final Position Latitude | Longitude | TD [m] |
|---|---|---|---|---|---|
| CAa | 39°55′40.51″ N | 18°02′56.23″ E | 39°55′40.56″ N | 18°02′56.26″ E | 1.6 |
| CAb | 39°55′40.55″ N | 18°02′56.25″ E | 39°55′40.60″ N | 18°02′56.27″ E | 1.8 |
| CAc | 39°55′40.48″ N | 18°02′56.22″ E | 39°55′40.60″ N | 18°02′56.20″ E | 3.9 |
| CAd | 39°55′40.52″ N | 18°02′56.01″ E | 39°55′40.56″ N | 18°02′56.10″ E | 2.5 |
| CAe | 39°55′41.15″ N | 18°02′55.35″ E | 39°55′41.25″ N | 18°02′55.36″ E | 3.2 |
| CAf | 39°55′41.62″ N | 18°02′54.82″ E | 39°55′41.66″ N | 18°02′54.89″ E | 1.9 |

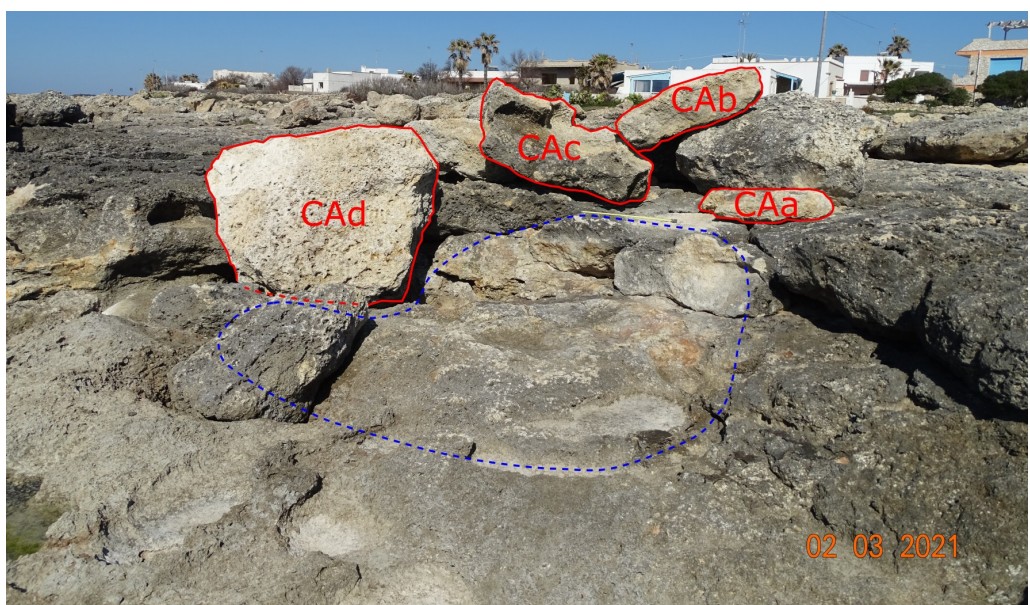

**Figure A6.** Capilungo coast. The boulder CAd and the group CAa,b,c taken from the coastline; the area of the initial positions is marked by the blue dashed line. Compare with Figure 9.

**Table A12.** Main features of the displaced boulders at the Capilungo coast; see the caption of Table A2 for symbols.

| Boulder ID | $a$ [m] | $b$ [m] | $c$ [m] | $x_i$ [m] | $x_f$ [m] | Li | Sh | FI | PTS | MT |
|:---:|:---:|:---:|:---:|:---:|:---:|:---:|:---:|:---:|:---:|:---:|
| CAa | 1.1 | 0.6 | 0.3 | 6.4 | 8.2 | C | B | 2.83 | SA | ST |
| CAb | 1.5 | 1.3 | 0.5 | 7.9 | 9.6 | C | D | 2.80 | SA | ST |
| CAc | 2.1 | 1.4 | 0.4 | 5.7 | 8.3 | C | D | 4.38 | SA | ST |
| CAd | 2.6 | 2.4 | 0.9 | 3.9 | 6.7 | C | D | 2.78 | SA | OV |
| CAe | 1.8 | 1.6 | 0.6 | 4.7 | 5.6 | C | D | 2.83 | SA | OV |
| CAf | 2.6 | 1.9 | 0.5 | 3.3 | 5.4 | C | D | 4.50 | SA | OV |

The last two surveyed sites (Ciardo and Torre Sant'Emiliano), although both with few boulders, are quite significant both for the lithology (limestone) and for the average height of the cliffs (2–3 m above the MSL). The transport of the four boulders in Figure A7 took place parallel to the coastline (SSE-NNW). Note the few altered surface of the sockets belonging to the CIa,b,c,d group. In the present study, the geometric centre of the overall detachment surface is considered as the initial position for this group (Tables A13 and A14). Greater accuracy can be achieved, for example, by using drone photogrammetric surveys to attribute each boulders to the own socket.

**Table A13.** Initial and final geographical coordinates of the displaced boulders at the Ciardo coast; ind., indeterminable.

| Boulder ID | Initial Position | | Final Position | | TD [m] |
|:---:|:---:|:---:|:---:|:---:|:---:|
| | Latitude | Longitude | Latitude | Longitude | |
| CIa | 39°48′09.50″ N | 18°19′42.41″ E | 39°48′09.51″ N | 18°19′42.29″ E | 2.8 |
| CIb | 39°48′09.50″ N | 18°19′42.41″ E | 39°48′09.53″ N | 18°19′42.35″ E | 1.7 |
| CIc | 39°48′09.50″ N | 18°19′42.41″ E | 39°48′09.58″ N | 18°19′42.31″ E | 3.3 |
| CId | 39°48′09.50″ N | 18°19′42.41″ E | 39°48′09.55″ N | 18°19′42.39″ E | 1.6 |
| CIe | 39°48′09.53″ N | 18°19′42.20″ E | ind. | ind. | - |

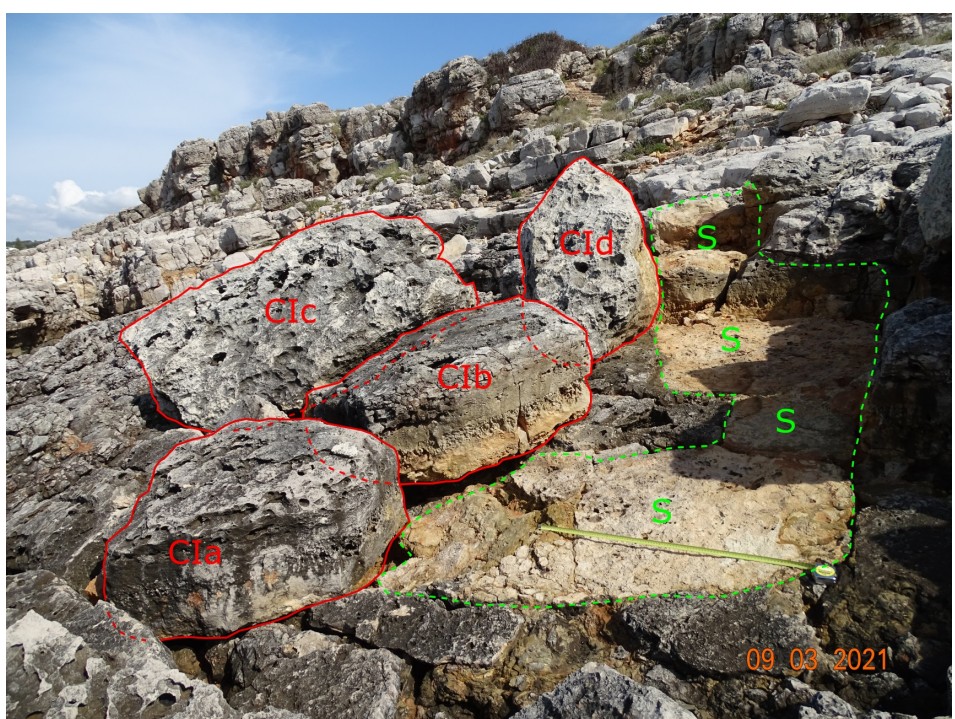

**Figure A7.** Ciardo coastal stretch. The boulders CIa, CIb, CIc, CId, and their sockets taken from the coastline.

**Table A14.** Main features of the displaced boulders at the Ciardo coast; see the caption of Table A2 for symbols.

| Boulder ID | $a$ [m] | $b$ [m] | $c$ [m] | $x_i$ [m] | $x_f$ [m] | Li | Sh | FI | PTS | MT |
|---|---|---|---|---|---|---|---|---|---|---|
| CIa | 1.0 | 0.8 | 0.5 | 5.2 | 3.5 | L | D | 1.80 | JB | ST |
| CIb | 1.4 | 1.2 | 0.5 | 5.2 | 4.6 | L | D | 2.60 | JB | ST |
| CIc | 2.2 | 1.4 | 0.5 | 5.2 | 5.0 | L | B | 3.60 | SA | ST |
| CId | 1.9 | 0.9 | 0.7 | 5.2 | 5.9 | L | P | 2.00 | JB | ST |
| CIe | 1.8 | 1.1 | ind. | 1.5 | ind. | L | - | - | JB | ind. |

The Ciardo and Torre Sant'Emiliano sites also have in common sockets clearly visible on GE (Figure A8). Scuba surveys are required to determine the post transport position of the boulders SEa,b (Figure A9).

**Table A15.** Initial and final geographical coordinates of the displaced boulders at the Torre Sant'Emiliano coast; ind., indeterminable. TD = Transport Distance (m).

| Boulder ID | Initial Position | | Final Position | | |
|---|---|---|---|---|---|
| | Latitude | Longitude | Latitude | Longitude | TD |
| SEa | 40°05′04.92″ N | 18°29′34.75″ E | ind. | ind. | - |
| SEb | 40°05′05.02″ N | 18°29′34.87″ E | ind. | ind. | - |

**Table A16.** Main features of the displaced boulders at the Torre Sant'Emiliano coast; see the caption of Table A2 for symbols.

| Boulder ID | $a$ [m] | $b$ [m] | $c$ [m] | $x_i$ [m] | $x_f$ [m] | Li | Sh | FI | PTS | MT |
|---|---|---|---|---|---|---|---|---|---|---|
| SEa | 6.7 | 3.8 | ind. | 3.6 | ind. | L | - | - | SA | ind. |
| SEb | 3.2 | 2.5 | ind. | 1.5 | ind. | L | - | - | SA | ind. |

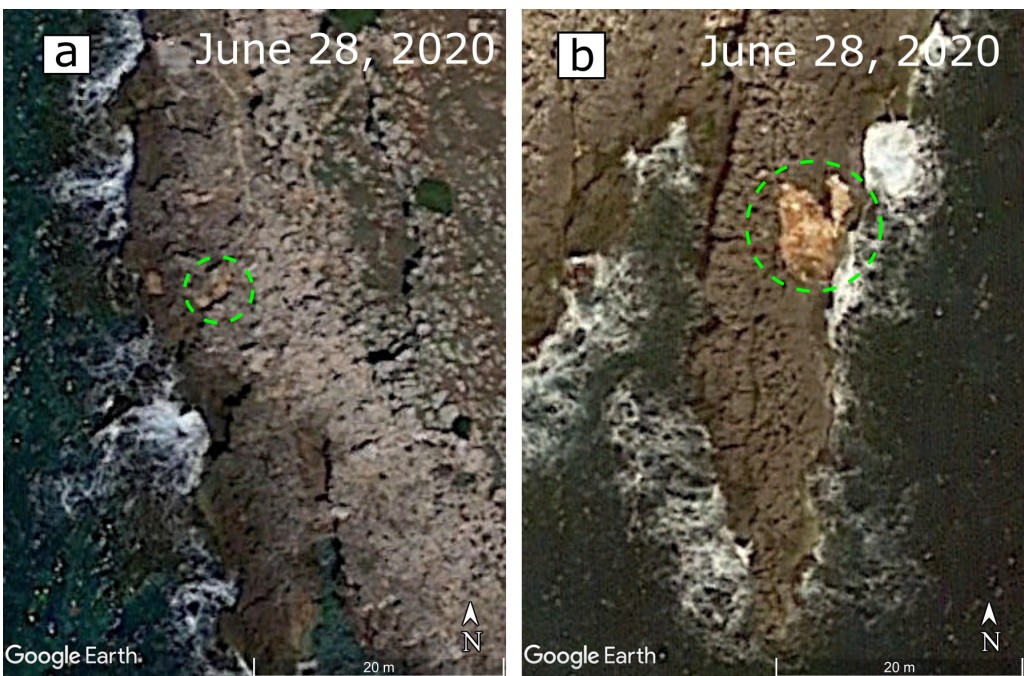

**Figure A8.** Satellite images of sockets: (**a**) The June 2020 image of the sockets of the boulders CIa,b,c,d (eye elevation of 35 m). (**b**) The June 2020 image of the sockets of the boulders SEa,b (eye elevation of 35 m).

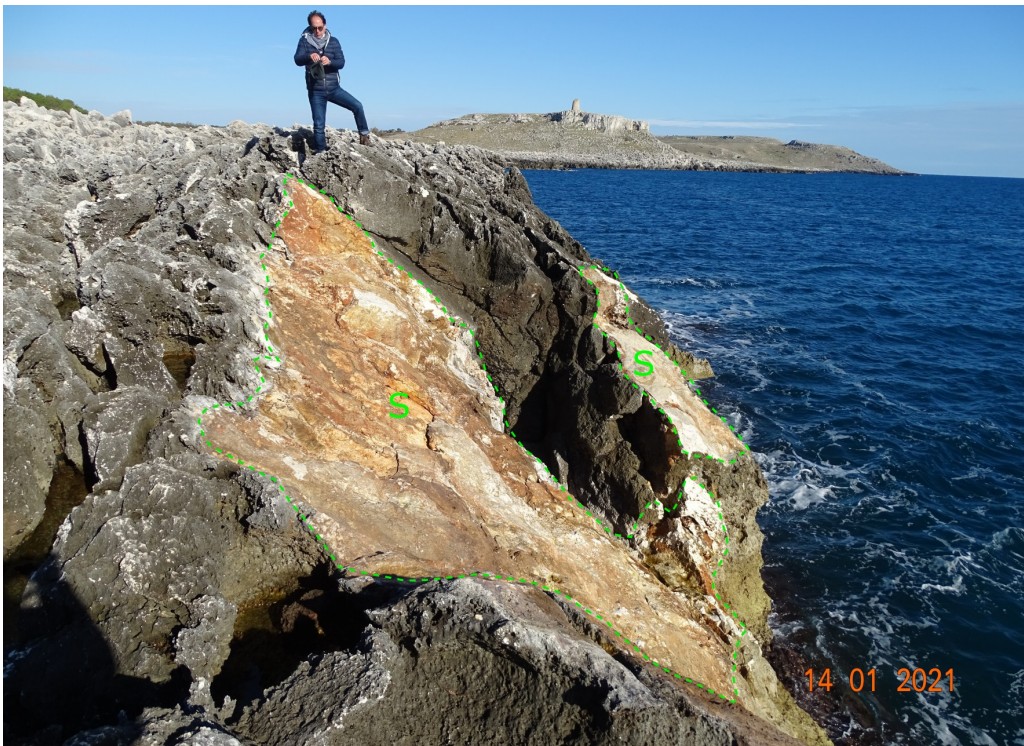

**Figure A9.** Torre Sant'Emiliano coast. The sockets of the boulders SEa and SEb are highlighted by the green dashed lines.

**Appendix B**

TD measuring and testing made at the central area of the Punta Pizzo headland (Section 4.4) are briefly described in what follows. The example was chosen given the wide range of TD in this site. Taking into account the procedure shown in Figure 2, for five out of eight displaced boulders (PIg, PIh, PIj, PIm, and PIn), both the initial and final positions

were recognized, and thus the TD was determined (Table A5). In Table A17, the difference between the measurements taken with the GE's ruler tool (from the 28 June 2020 GE image) and the tape measure (during the field survey, Figure A10) are shown.

**Table A17.** Difference between the measurements taken with the GE's ruler tool and the tape measure at the central area of Punta Pizzo headland.

| Boulder ID | GE Ruler [m] | Tape Measure [m] | Deviation |
|---|---|---|---|
| PIg | 2.8 | 3.0 | −0.2 |
| PIh | 14.3 | 13.9 | 0.4 |
| PIj | 15.6 | 15.3 | 0.3 |
| PIm | 0.6 | 0.6 | - |
| PIn | 0.7 | 0.8 | −0.1 |

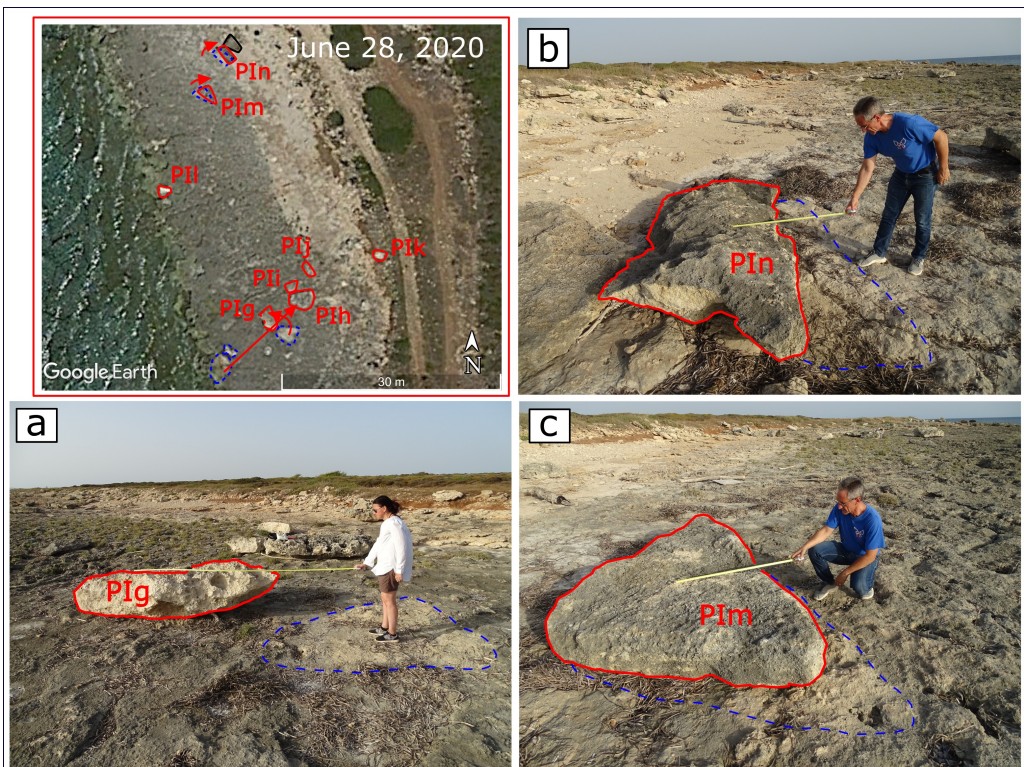

**Figure A10.** Central area of Punta Pizzo headland, examples of TD ground control (cf. Figure 5). Top left the 28 June 2020 GE image; (**a**) The tape is 3 m long. (**b,c**) The tape is 1 m long.

Similar results were obtained for all the sites herein considered. They are in agreement with the horizontal accuracy of GE images released in the last few years (see e.g., [66,67]). This leads to the conclusion that, within the study area, the TD measurements from the 28 June 2020 GE images are realistic.

**Appendix C**

The maps of the wind field at 10 m height for the storms described in Section 5 are shown below.

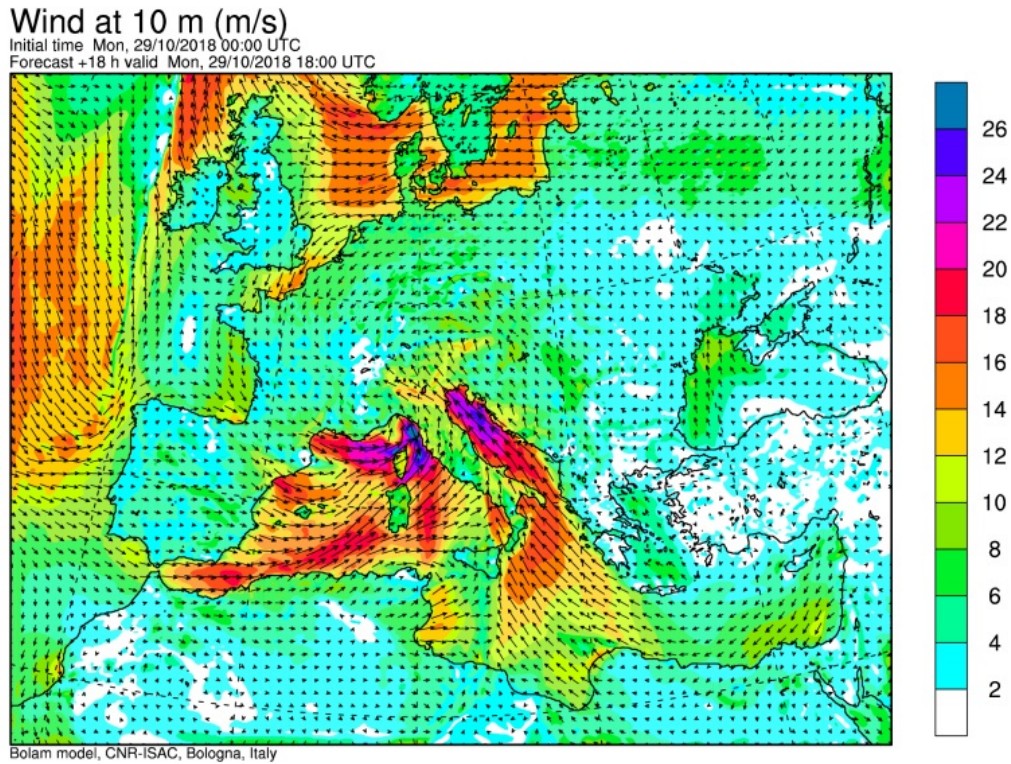

**Figure A11.** Map of the wind at 10 m height over the Mediterranean basin, as forecast for 29 October 2018 at 6 p.m. by the BOLAM model. The arrows indicate the wind direction and the wind speed is shown in the colour scale.

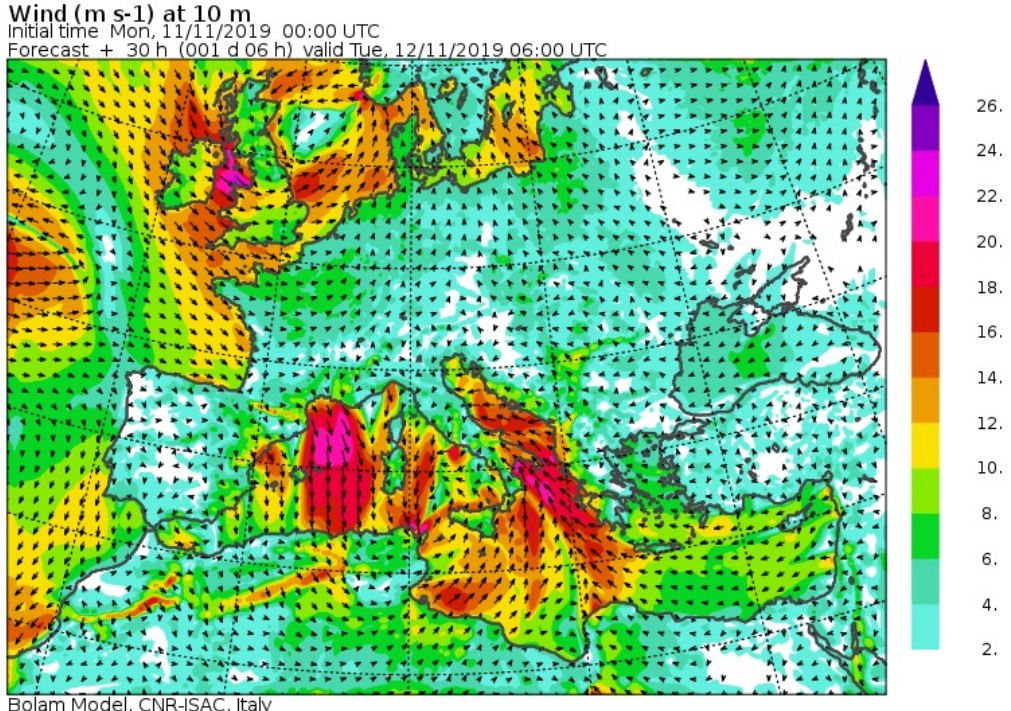

**Figure A12.** Map of the wind at 10 m height over the Mediterranean basin, as forecast for 12 November 2019 at 6 a.m. by the BOLAM model. The arrows indicate the wind direction and the wind speed is shown in the colour scale.

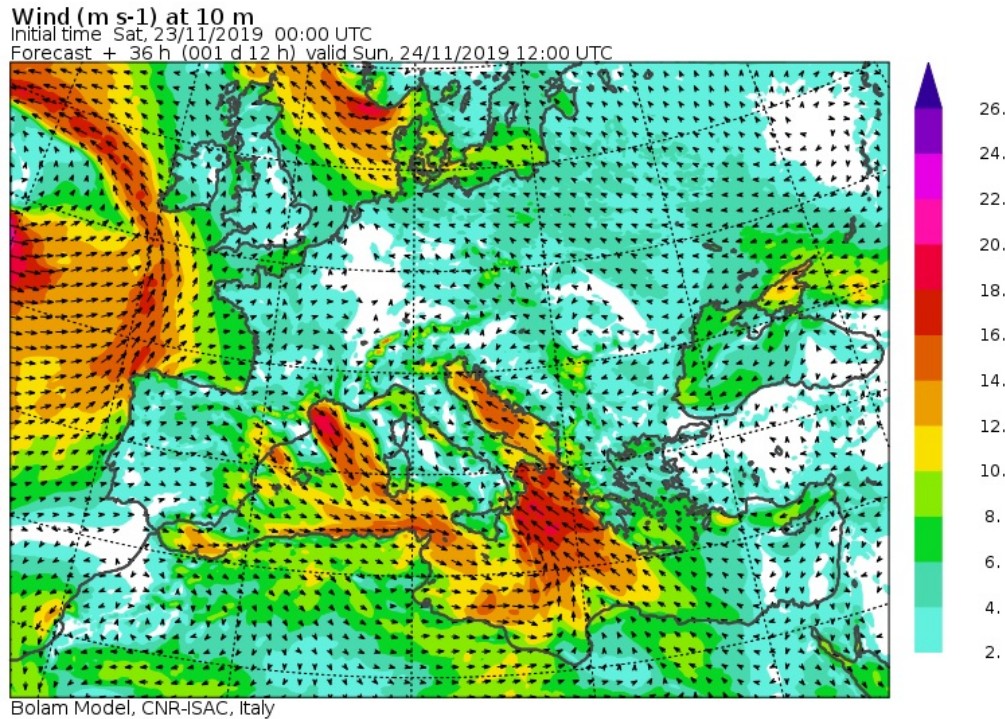

**Figure A13.** Map of the wind at 10 m height over the Mediterranean basin, as forecast for 24 November 2019 at 12 p.m. by the BOLAM model. The arrows indicate the wind direction and the wind speed is shown in the colour scale.

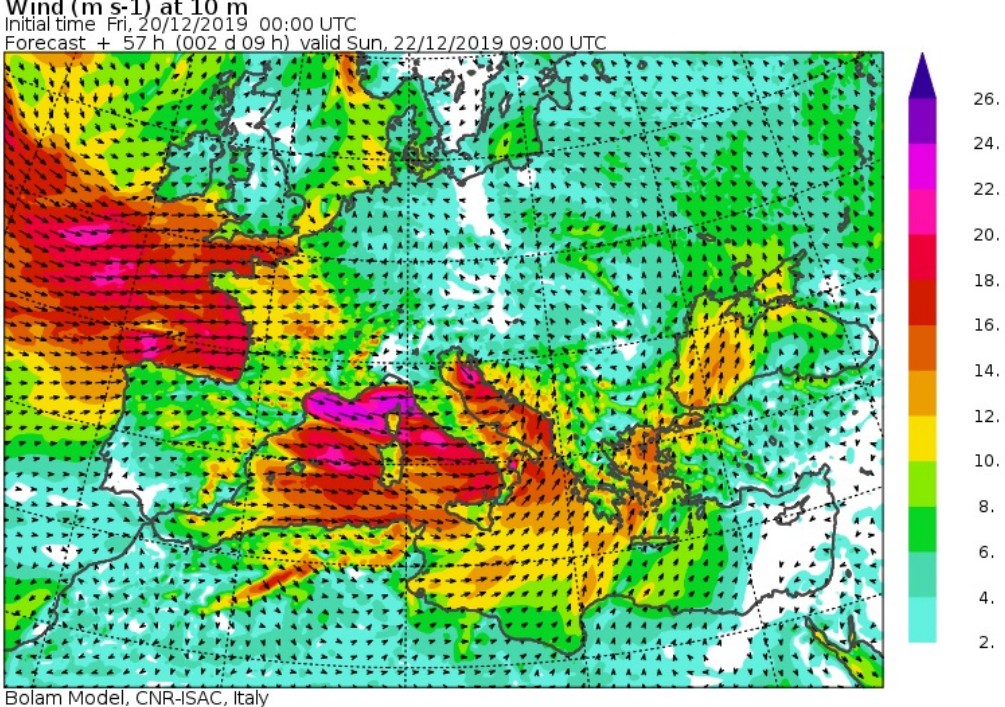

**Figure A14.** Map of the wind at 10 m height over the Mediterranean basin, as forecast for 22 December 2019 at 9 a.m. by the BOLAM model. The arrows indicate the wind direction and the wind speed is shown in the colour scale.

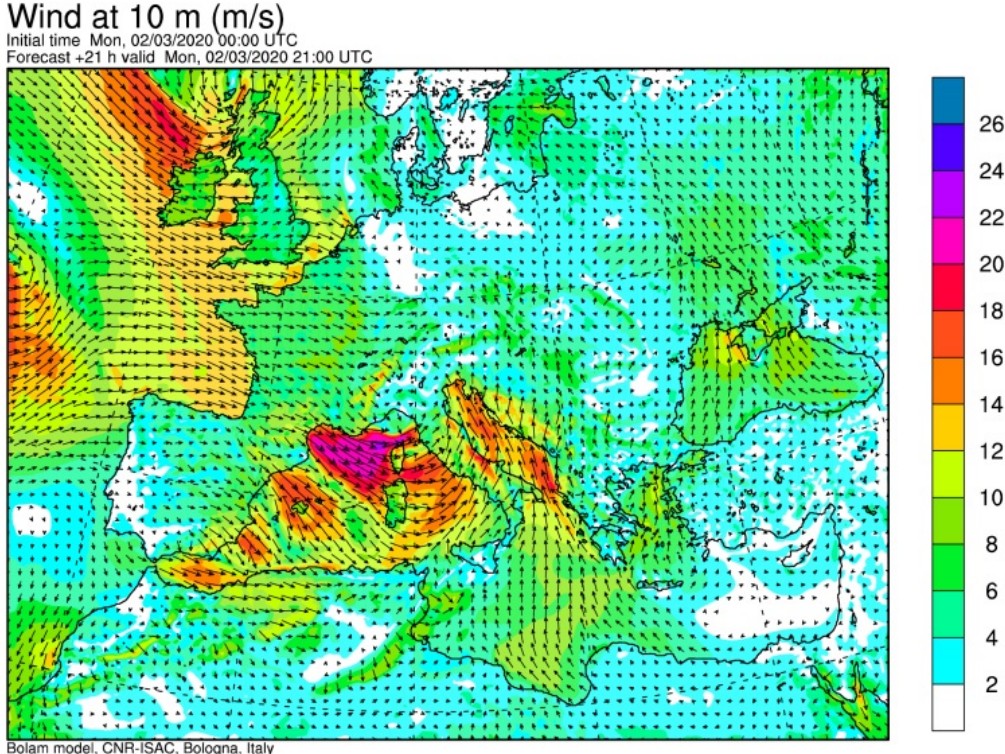

**Figure A15.** Map of the wind at 10 m height over the Mediterranean basin, as forecast for 2 March 2020 at 21 p.m. by the BOLAM model. The arrows indicate the wind direction and the wind speed is shown in the colour scale.

**Appendix D**

The hydrodynamic equations of [21,48] have been used to calculate the theoretical minimum height $H_m$ required to initiate the transport for the identified boulders by a sea wave. According to [21], for joint-bounded and sub-aerial boulders transported by saltation, $H_m$ is, respectively:

$$H_m = \frac{2c(\gamma_r/\gamma_w - 1)(\cos\theta + \mu\sin\theta)}{C_L} \tag{A1}$$

$$H_m = \frac{2c(\gamma_r/\gamma_w - 1)\cos\theta}{C_L} \tag{A2}$$

where $\gamma_r$ and $\gamma_w$ are the unit weights of rock and water, respectively, $\mu$ is the coefficient of static friction along rock surfaces, $\theta$ is the bed slope angle and $C_L$ is the lift coefficient. For sub-aerial boulders transported by sliding and by overturning, ref. [21] proposed, respectively:

$$H_m = \frac{2c(\gamma_r/\gamma_w - 1)(\mu\cos\theta + \sin\theta)}{C_D(c/b) + (\mu C_L)} \tag{A3}$$

$$H_m = \frac{2c(\gamma_r/\gamma_w - 1)(\cos\theta + (c/b)\sin\theta)}{C_D(c^2/b^2) + C_L} \tag{A4}$$

where $C_D$ is the drag coefficient.

Using a similar theoretical analysis [48], proposed a diagnostic equation for the limit case of boulders terminating their movement. Under the assumption that the last part of the motion is a sliding with negligible lifting and floating, they found:

$$H_m = \frac{2\mu V \gamma_r}{C_D(acq)\gamma_w} \tag{A5}$$

where $V$ is the boulder volume and $q$ is a boulder area coefficient that estimates the projected base-area. Estimating also the volume $V$ as $V = abcq$, which means the boulder height times projected base-area, Equation (A5) is equivalent to Equation (A3) with no floating, $C_L = 0$ and $\theta = 0$. We used Nandasena expressions (A1)–(A4) to test the starting condition for the boulder motion and expression (A5) to test the final condition. Literature data are used herein for the choice of the values of the coefficients in Equations (A3)–(A5); the selected values are: $\mu = 0.7$, $C_L = 0.178$, $C_D = 1.95$, $q = 0.73$ [7,21–23,48,57]. The bed slope angle $\theta$ is assumed to be zero due to the flat morphology of the study area.

The results are shown in Tables A18–A25.

In addition, the following equation [50] was used to estimate the incident wave height decrease over the shore, say the effective wave height $H$ impacting the boulder at a distance $X$ from the shoreline:

$$H = [(R + H_s - H_c)^{1/2} - 5X/(Tg^{1/2})]^2 \tag{A6}$$

In this equation, $T$ is the wave period, that can be estimated by Equation (1) $X$ the distance from the coastline and $g$ the gravity acceleration. The breaking wave height $R$ was directly estimated as its minimum value $H_0$ from Equation (2) (see also Section 6), and the average coastline height above the mean sea level $H_c$ was corrected by the total sea level increase (storm surge + tide) $H_s$ (= 1.5 m, see also Section 6.3).

**Table A18.** Minimumwave heights $H_m$ (m) required to displace the boulders of Punta Prosciutto headland.

| | Nandasena et al. (2011) [21] | Engel and May (2012) [48] |
|---|---|---|
| PRa | 3.6 | 1.2 |
| PRb | 7.2 | 1.4 |
| PRc | 4.5 | 1.7 |
| PRd | 1.6 | 1.6 |
| PRe | 1.1 | 1.5 |
| PRf | 1.3 | 1.4 |
| PRg | – | 1.3 |
| PRh | – | 1.6 |
| PRi | 3.6 | 1.8 |
| PRj | 2.7 | 1.1 |
| PRk | 3.6 | 1.8 |
| PRl | 1.6 | 1.4 |
| PRm | 1.1 | 1.1 |
| PRn | 4.5 | 2.9 |
| PRo | 3.6 | 1.6 |
| PRp | 5.4 | 1.8 |
| PRq | 5.4 | 1.7 |
| PRr | 3.6 | 1.7 |

**Table A19.** Minimum wave heights $H_m$ (m) required to displace the boulders of the Torre Sant'Isidoro coast.

| | Nandasena et al. (2011) [21] | Engel and May (2012) [48] |
|---|---|---|
| SIc | 0.7 | 2.0 |
| SIf | 3.6 | 2.9 |
| SIg | 2.0 | 1.8 |

**Table A20.** Minimum wave heights $H_m$ (m) required to displace the boulders of the Punta Pizzo headland.

|      | Nandasena et al. (2011) [21] | Engel and May (2012) [48] |
| ---- | ---------------------------- | ------------------------- |
| PIa  | 0.3                          | 0.8                       |
| PIb  | 0.5                          | 1.3                       |
| PIc  | 0.5                          | 1.3                       |
| PId  | 0.5                          | 1.3                       |
| PIe  | 0.4                          | 1.2                       |
| PIf  | 6.3                          | 1.2                       |
| PIg  | 1.7                          | 1.7                       |
| PIh  | 3.6                          | 2.6                       |
| PIi  | –                            | 1.1                       |
| PIj  | –                            | 0.8                       |
| PIk  | 0.8                          | 1.0                       |
| PIl  | –                            | 0.8                       |
| PIm  | 0.7                          | 2.0                       |
| PIn  | 0.5                          | 1.6                       |
| PIo  | 0.3                          | 0.7                       |
| PIp  | 2.0                          | 1.8                       |
| PIq  | –                            | 0.8                       |
| PIr  | 3.6                          | 1.7                       |

**Table A21.** Minimum wave heights $H_m$ (m) required to displace the boulders of the Mancaversa coast.

|      | Nandasena et al. (2011) [21] | Engel and May (2012) [48] |
| ---- | ---------------------------- | ------------------------- |
| MAa  | 4.5                          | 2.5                       |
| MAb  | 0.6                          | 1.8                       |
| MAc  | 1.5                          | 1.5                       |
| MAd  | –                            | 1.0                       |
| MAe  | –                            | 1.0                       |
| MAf  | 3.6                          | 0.6                       |
| MAg  | 2.7                          | 0.8                       |
| MAh  | 2.7                          | 0.6                       |
| MAi  | 2.7                          | 0.6                       |
| MAj  | 3.6                          | 1.8                       |
| MAk  | 4.5                          | 1.4                       |
| MAl  | 3.6                          | 1.4                       |
| MAm  | 0.7                          | 2.2                       |
| MAn  | 0.8                          | 2.4                       |
| MAo  | 1.4                          | 1.3                       |

**Table A22.** Minimum wave heights $H_m$ (m) required to displace the boulders of the Torre Suda coast.

| | Nandasena et al. (2011) [21] | Engel and May (2012) [48] |
|---|---|---|
| SUa | 2.0 | 5.5 |
| SUb | 4.5 | 1.0 |
| SUc | 6.3 | 1.1 |
| SUd | – | 3.8 |
| SUe | 0.7 | 2.2 |
| SUf | 4.5 | 2.3 |
| SUg | 4.5 | 1.3 |
| SUh | 7.9 | 1.7 |
| SUi | 0.4 | 1.1 |
| SUj | 0.4 | 1.2 |
| SUk | 0.8 | 2.0 |
| SUl | 1.2 | 3.4 |
| SUm | 5.4. | 1.7 |
| SUn | 0.8 | 2.3 |

**Table A23.** Minimum wave heights $H_m$ (m) required to displace the boulders of the Capilungo coast.

| | Nandasena et al. (2011) [21] | Engel and May (2012) [48] |
|---|---|---|
| CAa | 2.7 | 0.7 |
| CAb | 4.5 | 1.6 |
| CAc | 3.6 | 1.7 |
| CAd | 3.2 | 2.9 |
| CAe | 2.1 | 1.9 |
| CAf | 2.6 | 2.3 |

**Table A24.** Minimum wave heights $H_m$ (m) required to displace the boulders of the Ciardo coast.

| | Nandasena et al. (2011) [21] | Engel and May (2012) [48] |
|---|---|---|
| CIa | 6.7 | 1.2 |
| CIb | 6.7 | 1.8 |
| CIc | 6.7 | 2.1 |
| CId | 9.4 | 1.3 |
| CIe | – | 1.6 |

**Table A25.** Minimum wave heights $H_m$ (m) required to displace the boulders of the Torre Sant'Emiliano coast.

| | Nandasena et al. (2011) [21] | Engel and May (2012) [48] |
|---|---|---|
| SEa | – | 5.6 |
| SEb | – | 3.7 |

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
