# Peer review of "Coastal Boulder Dynamics Inferred from Multi-Temporal Satellite Imagery, Geological and Meteorological Investigations in Southern Apulia, Italy"

_water, doi:10.3390/w13172426_

Round 1
Reviewer 1 Report
I find the paper by M. Delle Rose et al. very interesting. They propose the use pot satellite free images to infer the dynamics of coastal boulders along the southern italian coast.
I find the paper worth publishing and interesting from two main points of view:
- The proposed methodology which can be applied by others and to monitor large areas with a limited use of resources;
- The possibility to identify the origin of single events.
This is interesting also because it shows that storm events (and not only tsunamis) can be the origin of the displacement of coarse and semi-coarse boulders. This is mostly relevant for the interpretation of past events and dynamics.
The paper is generally well written and structured, english is good (at least for a non native speaker as I am) and I do not have modifications to be suggested to the authors.
Author Response
Your comments: I find the paper by M. Delle Rose et al. very interesting. They propose the use pot satellite free images to infer the dynamics of coastal boulders along the southern Italian coast.
I find the paper worth publishing and interesting from two main points of view:
1. The proposed methodology which can be applied by others and to monitor large areas with a limited use of resources;
2. The possibility to identify the origin of single events.
This is interesting also because it shows that storm events (and not only tsunamis) can be the origin of the displacement of coarse and semi-coarse boulders. This is mostly relevant for the interpretation of past events and dynamics.
The paper is generally well written and structured, English is good (at least for a non native speaker as I am) and I do not have modifications to be suggested to the authors.
Reply: Thanks for the compliments. You have well grasped the aims of our research. As for the English, however, we tried to improve it in the revised version according to the comment of other reviewers.
Reviewer 2 Report
Reviewer:
The paper presents a study on the Coastal Boulder Dynamics Inferred from Multi-Temporal Satellite Imagery, Geological and Meteorological Investigations in Southern Apulia, Italy. The investigation aims gathering data on boulder displacements by using publicly available satellite imagery in order to identify the events of the change in the positions of the boulders according to the marine weather, to take boulders as a proxy for coastal hazard.
The topic is interesting, and the paper is in general very well written and accurate. There are however some minor points that should be clarified in my opinion before a possible publication as follows:
- Line 18: Can we mention here about INSAR method also?
- Line 19: What about satellite altimetry? Can we use altimetric observations or not? Why?
- Line 31: What about Sea level rise?
- Line 37: Could you please discuss here why your paper is going to investigate in comparison with the former studies?
- Line 120: You should rewrite this part.
- Line 167: I would suggest using acronym for Sea Wave Height (SWH) to avoid the reptation of this term.
- Line 296: Could you define MSL term? Before using it.
- Line 307: You have already provided acronym for Transport Distance (TD) here, why you don’t use only TD in the following sentences?
- Line: again, TD here.
Author Response
General comment: The paper presents a study on the Coastal Boulder Dynamics Inferred from Multi-Temporal Satellite Imagery, Geological and Meteorological Investigations in Southern Apulia, Italy. The investigation aims gathering data on boulder displacements by using publicly available satellite imagery in order to identify the events of the change in the positions of the boulders according to the marine weather, to take boulders as a proxy for coastal hazard. The topic is interesting, and the paper is in general very well written and accurate.
Reply: Thanks for your general comment. We have sent to the Water Office a copy of the revised manuscript highlighting the changes made.
Comment 1 - Line 18: Can we mention here about INSAR method also?
Reply: To the best of our knowledge, no papers have been published that used INSAR in coastal boulder monitoring.
Comment 2 - Line 19: What about satellite altimetry? Can we use altimetric observations or not? Why?
Reply: Thank you for such question. There are already some recent applications of Airborne LiDAR altimetry in literature (see for examples Autret et al 2018 [9 in References]). Currently, in coastal boulder studies satellite altimetry is used especially for long-term validation of wave height measurements.
Comment 3 - Line 31: What about Sea level rise?
Reply: The climatic sea level rise was not explicitly mentioned in the Introduction because not directly related to the topic of the study as its contribution to the boulders movement is expected to be small in the short-medium term. The global sea level increase by thermal expansion is if the order of few cm per °C of increasing global temperature, and the more relevant possible contribution due to the earth's ice melting would take a quite uncertain and possibly long time. Thus we limited almost completely to consider the possible climatic changes in the storms characteristics.
Comment 4 - Line 37: Could you please discuss here why your paper is going to investigate in comparison with the former studies?
Reply: The potentiality issue of Google Earth engine in costal boulder research is now exposed in Introduction.
Comment 5 - Line 120: You should rewrite this part.
Reply: Rewritten, thanks.
Comment 6 -Line 167: I would suggest using acronym for Sea Wave Height (SWH) to avoid the reptation of this term.
Reply: Thanks, suggestion accepted.
Comment 7 - Line 296: Could you define MSL term? Before using it.
Reply: MSL (Mean Sea Level) is defined at the line 72 of the first version (line 74 of the revised version).
Comment 8 - Line 307: You have already provided acronym for Transport Distance (TD) here, why you don’t use only TD in the following sentences? Line: again, TD here.
Reply: You are right. We have made the corrections.
Reviewer 3 Report
This paper presents research on a hot (and pressing topic) related to boulder movements during storms (or tsunamis) along developed coasts. I applaud the authors for taking on this research, however this work needs to be improved in several key areas.
My biggest hurdle with this paper is the lack of reporting on positional uncertainty. There are numerous papers out there related to measuring and estimating uncertainty from aerial photographs, particularly in the shoreline change literature. The authors are claiming to measure TD (transport distance, although units not reported in the tables) of less than a meter in some cases. This positional uncertainty is likely unrealistic given the source of the images and resolution of the images. I did a cursory analysis of some of the areas mentioned using Google Earth and features that have not likely moved (building corners, bridges etc.. ) show at minimum a meter or more of ‘shift’ between images in Google Earth (in some cases more). I am also not certain why they are relying on google earth rather than downloading the original satellite images?
It is not that I do not accept that the modeled storm waves could move the boulders, it is that to me, not showing error bars, +/- on positional accuracy etc. is a non-starter as far as publishing this type of work. This is not insurmountable however, and the authors have likely considered error, but it needs to be reported. If the positional uncertainty is higher than the measured change, can you say it has changed at all?
A few other issues:
- Validation with any wave buoy data during the actual storms they are linking to causation would be helpful
- Validation of water level/storm surge estimates from tide gauges? I understand the model estimates, but can these be validated?
- The wording/English is awkward in many places. This is more than style, in some places it has made the text a bit unclear. I have highlighted some of these in the attached document.
- The manuscript is written in a way that many paragraphs can be combined; one sentence is not a paragraph!
- Figure 1: Awkward colors here; the aqua blue looks like water. Suggest different colors and more detailed explanation than 1, 2, 3
- Line 108 – How are you defining ‘coast’
- There are numerous typos (see line 120 ‘Firtst’) and the editing needs a careful look.
- I suggest moving some of the field/site photographs into the text

Author Response
Comment 1: This paper presents research on a hot (and pressing topic) related to boulder movements during storms (or tsunamis) along developed coasts. I applaud the authors for taking on this research, however this work needs to be improved in several key areas.
Reply: Thank you for your general comment. Through your comments, we believe we have improved the paper in the key areas. In the revised manuscript, attention is given to the TD ground control and the related deviation. Please, for the revised manuscript with highlighted the changes made see the attached file.
Comment 2: My biggest hurdle with this paper is the lack of reporting on positional uncertainty. There are numerous papers out there related to measuring and estimating uncertainty from aerial photographs, particularly in the shoreline change literature. The authors are claiming to measure TD (transport distance, although units not reported in the tables) of less than a meter in some cases. This positional uncertainty is likely unrealistic given the source of the images and resolution of the images. I did a cursory analysis of some of the areas mentioned using Google Earth and features that have not likely moved (building corners, bridges etc…) show at minimum a meter or more of ‘shift’ between images in Google Earth (in some cases more). I am also not certain why they are relying on google earth rather than downloading the original satellite images?
Reply: We agree that the positional uncertainty is a crucial issue in several field of research such as shoreline change processed by satellite or aerial images. However, based on your comments, we understand that in the first version of our manuscript it is not well exposed how TD was obtained and ground controlled. As we explain in the revised manuscript (with the addition of the new Appendix B), initial and final positions of Table A1, A3, etc., are all taken on the June 28, 2020 satellite images, hence there is no the problem of "shift between images". Coupling imagery observations with geological field surveys (Section 1), for 59 out of 81 boulders (Section 4) both the positions were identified and, then, TD measured. The accuracy of TD is controlled by ground measure (see the new Figure 2 and Appendix B). Units (m) are now reported in the tables. Further explanations are below. Finally, we rely on Google Earth motivated by the literature on the open source use (Gorelik et al 2017; Kumar and Mutanga 2018; Ruban 2020, to quote a few), as better described in the revised manuscript.
Comment 3: It is not that I do not accept that the modeled storm waves could move the boulders, it is that to me, not showing error bars, +/- on positional accuracy etc. is a non-starter as far as publishing this type of work. This is not insurmountable however, and the authors have likely considered error, but it needs to be reported. If the positional uncertainty is higher than the measured change, can you say it has changed at all?
Reply: The comparison between the distances measured on the ground with those measured by Google Earth Engine, provides an initial evaluation of the errors that can be made (see the new Appendix B), giving the order of magnitude of the uncertainty of the TD measurements. On the ground, distance measurements were made using a tape measure. However, to achieve more reliable conclusions “detailed expensive studies on each of the considered sites must be carried out (i.e. photogrammetric, laser scan, scuba surveys)”, see lines 463-465 of the first version. On one of the surveyed sites, such investigations have begun and it is hoped that TD accuracy can be further improved.
Comment 4: Validation with any wave buoy data during the actual storms they are linking to causation would be helpful.
Reply: The data of the Crotone buoy (line 79 of the first version; line 81 of the revised version) of the Italian Ondametric Network are not available for the last seven years. We have planned to valid our inferences as well as data become publicly accessible. However, for the calculated spectral peak wave height H0, that is by far the major contribution to the total wave height H responsible for the boulders movement, a validation had been previously made with the data recorded by the meteomarine station of S Maria di Leuca (see map in Figure 1) for the same period [ref. 10]. This has been mentioned ad referenced in lines 357-358 of the old version (lines 365-366 of the revised version).
Comment 5: Validation of water level/storm surge estimates from tide gauges? I understand the model estimates, but can these be validated?
Reply: Unfortunately there are no tide gauges along the studied coast. The tide gauge closest to it, is located in the Taranto harbour (several tens of km to the northwest). However, evaluations are underway on the possible relationships between the sea water level recorded in the aforementioned port and the calculated storm surges. The tide contribution to the effective wave height (H) is very small and in agreement with the observed local diurnal tide.
Comment 6: The wording/English is awkward in many places. This is more than style, in some places it has made the text a bit unclear. I have highlighted some of these in the attached document.
Reply: lines 6-7, text edited; line 10, “high” deleted”; line 19, change made; line 25, paragraph deleted; lines 30-31, change made; line 32, text edited; lines 39-40, the sentence has been changed according to your recommendation; line 78, paragraph deleted; lines 78-79, text edited; line 85, text edited; line 120, text edited; first line of Table 1, “TD N.” changed with “No. of TD”; Line 186, text edited; lines 188-189, text edited; line 192, text edited; line 390, “an about” has been deleted and “long” has been changed with “of the Apulia”; lines 422-428 (fourth sticker comment on PDF: This is not a clear sentence. You are claiming these are not moved by a tsunami, but with the methods and lack of accuracy/precision reported you cannot make these claims. Even if you could, this paragraph is not clear and focused on your conclusions), paragraph deleted; line 522, “eighth-one” changed with “eighty-one”. In addition to your above recommendations, we have improved the language in other parts of the text.
Comment 7: The manuscript is written in a way that many paragraphs can be combined; one sentence is not a paragraph!
Reply: Several paragraphs have been combined.
Comment 8, Figure 1: Awkward colors here; the aqua blue looks like water. Suggest different colors and more detailed explanation than 1, 2, 3 (the same comment is in the first sticker of your PDF).
Reply: the colors of the geological units have been changed in accordance with international conventions; in the caption, the main lithology of each unit were added.
Comment 9: Line 107 – How are you defining ‘coast’ (the same comment is in the third sticker of your PDF).
Reply: The initial and final positions were measured from the “cliff edge”. We have amended the text to avoid misunderstanding.
Comment 10: There are numerous typos (see line 120 ‘Firtst’) and the editing needs a careful look.
Reply: We have corrected the typos. We hope to have found them all.
Comment 11: I suggest moving some of the field/site photographs into the text (see also the comment annotated next to Figure A1 of the first version).
Reply: we have moved two field/site photographs into the text (Figures 4 and 8 of the revised version).
Additional comments (reported on your PDF).
Comment 12, lines 47-51: you handwrite “Results”.
Reply: Results have been added in Introduction.
Comment 13, line 84: Are these all satellite? Some aerial?
Reply: We don't have aerial images. As a next step, a photogrammetric survey is planned.
Comment 14, second sticker: More detail needed to describe the methods here. Why Google Earth? Why not ArcMap? How was positional error and distortion from the images quantified?
Reply: see previous replies.
Comment 15, handwritten annotation (without uncertainty you can’t even say this!)
Reply: see previous replies.
Comment 16, handwritten annotations, Figures 7 and 8, Table A1 (errors bars? +- of position).
Reply: see previous replies.
Comment 17, handwritten annotation, lines 401-413 (too many short paragraphs).
Reply: The paragraphs have been combined.
Comment 18, handwritten annotation, Figure 9, (confusing figure).
Reply: The figure has been enlarged.

Reviewer 4 Report
My comments are included in the PDF as comment stickers. Please refer there to them.

Author Response
Dear Reviewer, in reviewing our manuscript, you have stated that: it provide sufficient background and include all relevant references (in introduction); the research design id appropriate; the methods are adequately described; the results are clearly presented; the conclusions are supported by the results.
Reply: Thanks for your approval. Please, for the revised manuscript with highlighted the changes made see the attached file.
Your 33 comments (as comment stickers) were included in a PDF. Here are our replies:
Comment 1 (lines 22-24): but you may need as well high-resolution satellite data but is it not in off to have aerial high-resolution images?
Reply: The use of Google Earth engine follows the literature that refers to publicy available satellite imagery as well as the open source use (Gorelik et al 2017; Kumar and Mutanga 2018; Ruban 2020, to quote a few). Each site verifications and insights must will use detailed expensive studies [see lines 463-466], including the processing of high-resolution images.
Comment 2 (Figure 1): the north arrow is missing.
Reply: The north arrow is now drawn.
Comment 3 (little box of Fig. 1): the picture is very general but there is no scale nor North Arrow helping the reader.
Reply: The north arrow and the scale are now drawn.
Comment 4 (line 53): belongs (NOT belong)
Reply: Correction done.
Comment 5 (line 73): where did these data come from (the isobaths along the coast) ? Could specifically add the reference both in the legend of figure 1 and in the text? Or is this a result from your own study?
Reply: Isobaths come from the nautical chart of "Istituto Idrografico della Marina Militare" of Italy. Reference was added, thanks.
Comment 6 (line 79): I can't see it on the map?
Reply: the position of the Crotone buoy has been added on the map.
Comment 7 (line 120): First (NOT Firtst).
Reply: Sorry for the typos.
Comment 8 (line 128): Instead of putting an entire book as a reference, it's better for the ease of the reader to specify the page where the reference supports the sentence or at least the chapter ? Something to check for each book cited in this paper!
Reply: For each equations, book and page are now indicated.
Comment 9 (line 151): I guess it is Fm and not Fs here?
Reply: You have right, thanks. Change made.
Comment 10 (line 184): space missing
Reply: Correction done.
Comment 11 (caption of Table 1): in this abbreviation, N stands for what?
Reply: “N” stood for “number of”. With your comment, we have found that the wording is not right. In the revised version, the reader found “No. of”.
Comment 12 (line 188): spaces missing see P. and T.
Reply: Corrections done.
Comment 13 (Figure 2): no North arrow?
Reply: The north arrow is now drawn.
Comment 14 (line 210): "the first group" instead of the former
Reply: Corrections done.
Comment 15 (line 211): blue (NOT blu)
Reply: Correction done.
Comment 16 (line 212): because of what, they come from a submerged part of the shelf, they look overturned so they should originate from a closer source area? at the same time, if you look to the satellite images, at the east of the top blue boulder -> the situation/position of the boulders are not the same either (look at the place just above the red boulders PRnop ? any comment about that?
Reply: There are no traces of activity of marine organisms on these boulders, thus a provenance from a submerged part of the shelf may be excluded. PRnop are probably overturned but currently we have not data to infer their initial positions. Again, you are right about the change in positions of boulders at the east of the top blue boulder. These are problems to be solved with a thorough study of the site (see Section 6).
Comment 17 (line 220): remove "the" or add "the year 2017"
Reply: Correction done.
Comment 18 (Figure 3): add north arrow on the map
Reply: The north arrow is now drawn. We have added arrow also on the Figures 5, 6, 7 and 9.
Comment 19 (line 265): check space before They
Reply: Check made (line 274 of the revised manuscript), only one space founded.
Comment 20 (line 291): lies (NOT lie)
Reply: Correction done.
Comment 21 (line 297): limestones (NOT limestone)
Reply: Correction done.
Comment 22 (line 315): these are the major storms or is it all the storms recorded? Do you have any ideas on the frequency of this recurrent storm activity? do you have any information for smaller storms events affecting also the coast even locally
Reply: As mentioned at the beginning of section 3.2 the stormy condition over the Mediterranean sea were found by a careful screening of the GLOBO 500 hPa geopotential height maps archive. This procedure ensures the proper identifications of all the stormy periods over the Mediterranean region, and to allow a subsequent more detailed analysis just for the identified storms. We cannot provide an estimate of the storm frequency over the area that should be properly classified by their strength, but the storms in the study period seem to be quite unusual for number and strength if compared with the preceding and following year.
Comment 23 (line 316): characteristics (NOT characteristic)
Reply: Correction done.
Comment 24 (line 380): space missing 10 m
Reply: space added, thanks.
Comment 25 (caption of Figure 7): I'm surprised by the scale which is created inversely to the size of the boulder .. this is a bit confusing for the reader! Why not taking the same philosophy both in size and in symbols?
Reply: The scale is not created inversely to the size of the boulder. Probably the misunderstanding is caused by the nomenclature of Blair and Mcpherson (1999): coarse boulder; very coarse boulder; fine block. Please note that the first and the second are qualified “boulder”, instead the third, the larger, “block”.
Comment 26 (line 425): I do not understand how the storms can modify the sedimentological structure of the boulder?
Reply: The storms cannot clearly modify the sedimentological features of a single boulder (which are diagenized and lithified). Instead they can modify the sedimentology of the “boulder accumulations”, which have to intended as groups of boulders, imbricated boulders, boulder ridges, and so on. However, the sentence causes misunderstanding. As a matter of fact, another reviewer has convinced us to remove the whole paragraph because of is not focused on our conclusions.
Comment 27 (line 465): I do believe that you take into account also the nature of the rocks as the density could be different and so more easy to transport if lighter compared to other type of rocks (a weathered limestone is not the same compared to a sandstone or even to a granite). How the nature of the rocks is taken into consideration from previous studies and in your study?
Reply: From previous studies and in our study, the nature of the rocks is taken into consideration. As we exposed in manuscript, three types of rocks have generated boulders: calcarenite (C), limestone (L), sandstone (S), see also caption of Table A2. By applying the equations in Appendix D, we have used the relative densities.
Comment 28 (line 522): eighty-one (NOT eighth-one)
Reply: Correction done.
Comment 29 (line 533): missing term here?
Reply: Text has been amended.
Comment 30 (line 563): sea-level rise
Reply: Correction done.
Comment 31 (Reference n. 6): year in bold
Reply: for Books and Book Chapters, the instruction for authors of Water reports the follow example: Author 1, A.; Author 2, B. Book Title, 3rd ed.; Publisher: Publisher Location, Country, Year; pp. 154–196. Please note that, unlike papers, year is not in bold.
Comment 32 (Reference n. 36): year in bold and check the other references as well to homogenize the reference style.
Reply: please see above (reply to your comment 31).
Comment 33 (Reference n. 40): when? no date
Reply: date has been added.

Round 2
Reviewer 3 Report
I am fine with this updated version, however some of the english/wording is still awkward (i.e. Line 227 'is monitored since the 2017' is very awkward).
Sorry for the delay; we were without power for a couple of days as the result of a storm.